# ON THE DYNAMICS OF COHERENT MEMORY STRUCTURES IN NEURAL FIELDS

## ABSTRACT

Memory in biological neural networks is often supported by coherent spatiotemporal patterns, such as traveling waves and neural activity confined to low dimensional manifolds, which are captured at mesoscopic scales by continuum neural field models. Substantial progress has been made in mechanistically analyzing both biological and artificial neural network architectures. Recent works obtain interpretable latent states by imposing traveling waves or low-dimensional invariant manifolds, but typically do not provide data-driven explanations for when and why such structures emerge during task training. We develop a theoretical framework for studying latent dynamics based on the Mori-Zwanzig projection-operator formalism. Our approach casts memory as a family of time-dependent projections that reveal how coupled dynamics support memory encoding and decoding. We instantiate a neural-field-inspired architecture, and evaluate it on both long-range benchmarks and neuroscience applications involving EEG and ECoG prediction. Across these tasks, we observe robust long-range accuracy and interpretable memory modes in the learned latent dynamics.

## 1 INTRODUCTION

Biological neural networks exhibit a range of coherent dynamical phenomena–such as stable attractor states and traveling waves (Engel et al., 2001; Wang, 1999; Engel & Steinmetz, 2019)–that are increasingly implicated in working memory and large-scale coordination across cortical regions. These phenomena have been studied using a variety of dynamical modeling frameworks, including neural field models (Ermentrout, 1998; Coombes, 2005a)–continuous, spatially extended dynamical systems that describe mesoscopic activity of densely interconnected neuron populations–as well as methods that reduce the neural dynamics to low-dimensional manifolds (Marrouch et al., 2020). These models provide mechanistic insight into how coherent population activity supports cognition, and have been used to explain core functions including the integration of sensory input, consolidation of long-term memories, and the organization of decisions, motor actions, and temporal sequences (Muller et al., 2014; Massimini et al., 2004; Rubino et al., 2006; Wimmer et al., 2014). These same attractor and wave phenomena are emergent in artificial neural networks (ANNs) (Rajan et al., 2016; Karuvally et al., 2024), and have been explicitly manipulated through architectural design to improve memory retention, sequence processing, and structured computation (Hopfield, 1982; Rusch & Mishra, 2021; Keller et al., 2024). Many of these architectures draw direct inspiration from biological neural networks, using insights from cortical dynamics to guide the design of memory and sequence-processing mechanisms in artificial systems. Some studies have attempted to bridge these motifs, demonstrating waves emerging from attractor instabilities (Coombes, 2005a) or attractor basins organizing wave propagation (Laing & Chow, 2002). However, they are typically studied in isolation (Ságodi et al., 2024; Karuvally et al., 2024; Keller et al., 2024), and a unifying theoretical framework to explain the flow of information in neural systems remains absent (Liu et al., 2025; Lei et al., 2024). Developing a framework that reconciles stability, propagation, and recall in biological and artificial networks remains a key challenge to explaining the dynamical behavior of intelligence (Alamia et al., 2025; Keller, 2025).

A central goal in neuroscience is to understand how cognitive functions emerge from complex neural activity, and dynamical models, particularly those grounded in attractor dynamics and traveling waves have proven essential for this purpose. Fixed-point and stationary bump attractors have been shown to stabilize persistent activity patterns in working memory (Wang, 1999; Wimmer et al.,

2014). Sequential and metastable attractors govern transitions between internal states, modeling decisions, motor plans, and temporal sequences (Friston, 1997; Kelso, 2012). Stochastic attractor exploration during rest and sleep facilitates internal simulation, cognitive flexibility, and memory consolidation through spontaneous traversal of neural state space (Deco et al., 2009; Chaudhuri et al., 2019). Traveling waves appear to serve complementary roles in coordinating activity across space and time, and in some instances emerge as instabilities of attractor states. Wavefronts, often stimulus-evoked (Muller et al., 2014) or multistability-driven (Laing & Chow, 2002), support sensory integration (Muller et al.), attentional shifts (Maris et al., 2013), and large-scale coordination (Takahashi et al., 2011) by propagating sustained activation through cortical maps. Wave pulses are localized, transient bursts, shaped by excitation-inhibition balance (Brunel, 2000) or excitability thresholds (Douglas et al., 1995), and are implicated in timing, signal relay, and motor planning (Rubino et al., 2006; Latash et al., 2010). Spontaneous waves emerge endogenously during anesthesia (Townsend et al., 2015), sleep (Massimini et al., 2004), or perception (Davis et al., 2020), traverse cortical attractor landscapes to support memory consolidation (Lee & Wilson, 2002), internal simulation, and synaptic refinement (Feller, 1999). Attractor dynamics and traveling waves are tightly linked in pattern-formation and continuum neural field models (Coombes, 2005b; Bressloff, 2013). Here we instead consider high-dimensional neural systems trained for specific computations, where attractor-based working-memory models and traveling-wave models are typically studied within separate frameworks. For such learned systems, it is still unclear how changes in the underlying dynamics reorganize memory representations over time (Liu et al., 2025).

Recurrent neural networks (RNNs) often struggle to retain information over long timescales due to vanishing/exploding gradients (EVGP) and the compression of long histories into finite-dimensional hiden states Bengio et al. (1994); Pascanu et al. (2013); attention mitigates the fixed-context bottleneck Bahdanau et al. (2014), even as EVGP remains an issue (Zucchet & Orvieto, 2024). Recent successes in deep state-space models (Gu & Dao, 2024) and transformer (Vaswani et al., 2017) architectures have overcome these challenges through structured state updates and self-attention mechanisms, respectively. However, even state-of-the-art transformers and deep state-space models can struggle with long-range dependencies and structured sequence tasks (Jelassi et al., 2024), highlighting the importance of understanding memory mechanisms. Inspired by biological systems, many RNNs have been imbued with (stable) attractor-like (Rusch & Mishra, 2021; Keller & Welling, 2023; Ságodi et al., 2024) or wave-like (Keller et al., 2024; Keller, 2025; Liu et al., 2025) structures to bolster memory retention and sequence processing. Intriguingly, even standard RNNs trained on history-dependent dynamical systems reveal latent waves under coordinate transforms (Karuvally et al., 2024). Inspired by neural fields, researchers have extended these ideas to practical applications, emulating cortical wave propagation for image segmentation (Liboni et al., 2025), modeling spatially working memory geometries (Lei et al., 2024) and sensory input (Xie et al., 2022).

Mori-Zwanzig (MZ) formalism offers an exact decomposition of a dynamical system into an equation over chosen variables, that explicitly accounts for the memory effects that shape their future behavior (Mori, 1965b; Zwanzig, 1961; Nakajima, 1958). Classical MZ is a technique developed for statistical mechanics that has been used to study molecular dynamics (Meyer et al., 2017), viscous Burgers flows (Stinis, 2012), and the Euler equations (Stinis, 2007). Data-driven machine learning approaches using MZ (Chorin et al., 2002; Lin et al., 2021; 2023) are a bottom-up approach to reduced-order modeling (Givon et al., 2004; Gupta et al., 2024) similar to time-delay embeddings (Woodward et al., 2025), which have shown recent success in modeling isotropic turbulence (Tian et al., 2021) and hypersonic boundary layer transitions (Woodward, 2023). More recently, MZ has been used as a framework for deep learning (Venturi & Li, 2023), where it has been used to inform the latent state of LSTMS (Maulik et al., 2020), as an effective auto-encoder (Gupta et al., 2024), to predict time-dependent PDEs using neural operators (Buitrago et al., 2025), and to enhance the explainability of neural networks (Menier et al., 2023). However, two assumptions made by MZ inspired deep learning architectures oversimplify the latent dynamics. First, many MZ architectures formulate memory using a time-delay of the latent state, neglecting the inclusion of the generalized fluctuation-dissipation relation (GFDR) in the memory kernel (Lin et al., 2023). Second, many MZ inspired architectures assume an at equilibrium state for the latent dynamics neglecting the effects of time-dependent memory kernels (Grabert, 2006; Héry & Netz, 2024; Netz, 2024; Venturi & Li, 2023). Moreover, the approaches that properly assume the structure of the latent dynamics neglect to account for the additional degrees of freedom often introduced during the encoding of information into the latent state. This approach is critical for learning time varying be-

havior of information in the latent state, where the latent state itself contains an over-representation of information. Recent work has linked these dynamics to the ability of data-driven MZ to discover emergent organization (Rupe & Crutchfield, 2024). To our knowledge, prior MZ-based approaches do not explicitly track how neural dynamics reorganize in latent space over the course of learning.

## 1.1 OUR CONTRIBUTION.

We present a novel theoretical framework for modeling the time-dependent dynamics of latent representations of an ANN during sequence learning. In particular:

1. We derive a generalized Langevin equation that accounts for intrinsic degrees of freedom using a family of time-dependent projections.
2. We provide practical guidance by implementing a biologically-inspired Neural Wave Field architecture equipped with MZ dynamics. By considering wave and oscillatory dynamics we are able to study information encoding and retrieval most naturally tied to the brain.
3. We empirically validate our approach by evaluating it on several long-range learning benchmarks and real-world neuroscience applications. We observe robust long-range recall, minimal memory dimension, and interpretable latent modes.

## 1.2 RELATED WORK

Several deep learning architectures using MZ-inspired time-delay memory, e.g. in neural operators and autoencoders (Buitrago et al., 2025; Gupta et al., 2024), do not explicitly enforce GFDR consistency as discussed in (Lin et al., 2023). Some data-driven techniques do enforce GFDR, e.g. through iterative regression (Lin et al., 2023; 2021). However, these approaches do not focus on coherent behaviors. Neural oscillators and traveling-wave networks directly encode coherent latent dynamics, but lack theoretical justifications for their dynamic behavior (Rusch & Rus, 2025; Keller et al., 2024). Transformers and structured state-space-models are often deployed as high-capacity predictors optimized for task performance (Gu et al., 2022; Fu et al., 2023; Gu & Dao, 2024).

By contrast, our approach aims to leverage coherent latent dynamics and enhance their mechanistic interpretability. This approach enables the model to suppress uninformative latents, elevate coherent structure, and shorten effective memory. See Appendix A for additional related works.

## 2 BACKGROUND

We first motivate our mathematical approach using a traveling wave example. We then formalize the background projection operator theory. Finally, we return to the traveling wave example to introduce notions of invariant-trivialization and projection-induced coherence.

## 2.1 MOTIVATION: COMPRESSING TRAVELING WAVE INFORMATION

To motivate our work, we consider a traveling wave over a *neural field*, which is a coarse-grained representation of cortex. Consider the field $u : [0, L] \times [0, T] \mapsto \mathbb{R}$ with traveling wave dynamics

$$\partial_t u(x,t) = -\nu \partial_x u(x,t) = \mathcal{S}u(x,t), \quad \nu > 0, \quad 0 \leq x \leq L, \quad u(0,t) = f(t),$$

where $\mathcal{S}$ is an advection (shift) operator implementing left-to-right transport at rate $\nu$ and the right boundary is an outflow. Information is injected at $x = 0$ by $f(t)$ and then travels across the field without re-entering or reflecting. We have chosen this input/free-flow configuration in contrast to prior works Keller et al. (2024) that impose periodic boundary conditions; see Appendix B.1 for a detailed discussion. This choice is also biologically motivated. For example, visual cortical input is injected at specific input layers and then flows feedforward through downstream populations.

Our aim is to study the compression of information in the latent state $u(x, \cdot)$ by restricting our view to a subset of the domain $z \subsetneq [0, L]$. After spatial discretization, we write the full state as $\boldsymbol{u}(t) \in \mathbb{R}^N$, let $\boldsymbol{v}(t) \in \mathbb{R}^m$ denote the restriction of $\boldsymbol{u}(t)$ to the *resolved* region $z$, and let $\boldsymbol{w}(t) \in \mathbb{R}^{N-m}$ denote the remaining *unresolved* components. We define a projection $P$ onto the resolved subspace by $P\boldsymbol{u} = (\boldsymbol{v}, 0)$ and $Q\boldsymbol{u} = (I - P)\boldsymbol{u} = (0, \boldsymbol{w})$. The dynamics of $\boldsymbol{u}(t)$ can be exactly decomposed as

$$\frac{d}{dt}\boldsymbol{u}(t) = \begin{pmatrix} P\mathcal{S}P & P\mathcal{S}Q \\ Q\mathcal{S}P & Q\mathcal{S}Q \end{pmatrix} \boldsymbol{u}(x,t) + \begin{pmatrix} P\boldsymbol{e}_0 \\ Q\boldsymbol{e}_0 \end{pmatrix} f(t) = \begin{pmatrix} S_{vv} & S_{vw} \\ S_{wv} & S_{ww} \end{pmatrix} \begin{pmatrix} \boldsymbol{v}(t) \\ \boldsymbol{w}(t) \end{pmatrix} + \begin{pmatrix} \boldsymbol{b}_v \\ \boldsymbol{b}_w \end{pmatrix} f(t) = \frac{d}{dt} \begin{pmatrix} \boldsymbol{v}(t) \\ \boldsymbol{w}(t) \end{pmatrix},$$

where the blocks $\boldsymbol{S_{vv}}, \boldsymbol{S_{vw}}, \boldsymbol{S_{wv}}, \boldsymbol{S_{ww}}$ are induced by the restriction of $\mathcal{S}$ to the resolved and unresolved coordinates.

The dynamics of $\boldsymbol{v}(t)$ are then described by the following non-Markovian system

$$\frac{d}{dt}\boldsymbol{v}(t) = \boldsymbol{S_{vv}}\boldsymbol{v}(t) + \int_0^t \boldsymbol{S_{vw}}e^{(t-s)\boldsymbol{S_{ww}}}\boldsymbol{S_{wv}}\boldsymbol{v}(s)\,ds + \boldsymbol{S_{vw}}e^{t\boldsymbol{S_{ww}}}\boldsymbol{w}(0) + \int_0^t \boldsymbol{S_{vw}}e^{(t-s)\boldsymbol{S_{ww}}}\boldsymbol{b_w}f(s)\,ds + \boldsymbol{b_v}f(t),$$

see Appendix C.5 for a derivation. Thus, even though the full system $\boldsymbol{u}(t)$ evolves Markovianly under $\mathcal{S}$, the compressed latent $\boldsymbol{v}(t)$ is non-Markovian.

**Lifting Information**  To remove the dependence on the boundary we will introduce a lifting operator that allows us to simplify the dynamics of $v$. In particular, we will assume (A1) that we trade temporal encoding of information on the boundary for spatially-resolved information at some time $k$. We do this by introducing $\mathcal{L}_k : (u(0), f_{[0,k]}) \to u(k)$. Then starting from time $k$ for $\tau = t + k$

$$\frac{d}{dt}\boldsymbol{v}(\tau) = \boldsymbol{S_{vv}}\boldsymbol{v}(\tau) + \int_k^\tau \boldsymbol{S_{vw}}e^{(\tau-s)\boldsymbol{S_{ww}}}\boldsymbol{S_{wv}}\boldsymbol{v}(s)\,ds + \boldsymbol{S_{vw}}e^{\tau \boldsymbol{S_{ww}}}\boldsymbol{w}(k).$$

With this non-Markovian system in hand, we reflect on the generalization of this technique before revisiting this example.

## 2.2 PROJECTION OPERATOR FORMALISM

This technique is formalized for a fixed choice of resolved *measurement* (i.e., representation of information) by near-equilibrium MZ (NE-MZ) (Mori, 1965a; Zwanzig, 2001). An extension to dynamic representations is given by the far-from-equilibrium MZ (FFE-MZ) (Grabert, 2006).

**Near-equilibrium**  The NE-MZ formalism provides an exact decomposition for the evolution of a measurement $\partial_t g = \mathcal{L}g$ with *resolved* $\hat{g} = Pg$ and *unresolved* $\tilde{g} = Qg$ components. The resolved evolution is given by the generalized Langevin equation (GLE)

$$\frac{\partial}{\partial t}\hat{g}(t) = \underbrace{P\mathcal{L}\,\hat{g}(t)}_{\text{Markov}} + \underbrace{\int_0^t P\mathcal{L}\,e^{(t-s)Q\mathcal{L}}\,Q\mathcal{L}\,\hat{g}(s)\,\mathrm{d}s}_{\text{Memory}} + \underbrace{P\mathcal{L}e^{tQ\mathcal{L}}\,Q\,g(0)}_{\text{Fluctuating Force}}. \tag{1}$$

Equation 1 consists of three distinct terms (underscored). The Markov term represents the instantaneous drift from the resolved dynamics. The Memory term re-introduces the influence of dynamics previously *forgotten*, i.e. prior resolved information that has been projected into the unresolved subspace. The Fluctuating Force term[1] captures the residual influence of the unresolved initial state.

**Far-from-equilibrium**  For a neural network architecture, it may not be possible, and potentially unreasonable to ascribe to each element of the latent state a static representation of *what* information is encoded. FFE-MZ (Grabert, 2006) is a prior approach to handling time-dependent measurements by introducing time-dependent $P(t)$. Suppose $P(t)$ is differentiable (A2), then the resulting GLE is

$$\frac{\partial}{\partial t}\hat{g}(t) = P(t)\mathcal{L}\hat{g}(t) + \dot{P}(t)g(t) + \int_0^t P(t)\mathcal{L}G(t,s)Q(s)\mathcal{L}\hat{g}(s)\mathrm{d}s + P(t)\mathcal{L}G(t,0)Q(0)\,g(0). \tag{2}$$

The two-time memory kernel $G(t,s) = \mathcal{T}_- \exp\left(\int_s^t Q(u)\,\mathcal{L}\,\mathrm{d}u\right)$ is the negatively time-ordered exponential operator that captures the *extrinsic* influence from the evolution of the subspaces. The Kinematic term $\dot{P}(t)g(t)$ and captures the intrinsic evolution of the resolved subspace (Meyer et al., 2017). The take-away is that the time-dependent projection operator acts as a moving frame of reference tied to the desired measurement.

Our approach is distinguished from FFE-MZ in that we do not obtain a two-time memory kernel; although, we derive a similar drift term. This is because our lift operator assumption allows us to suppose that the information of interest is intrinsically in the latent state. As a result, we do not recover extrinsic influences from the evolution of the subspaces. We leave extensions to partially observed systems and lift-free derivations to future work.

---

[1]The third term referred to by (Mori, 1965a) as a random force and by (Zwanzig, 2001) as a fluctuating force, is frequently called the noise term in data-driven and stochastic applications.

## 2.3 COHERENT DYNAMICS IN A COMPRESSED TRAVELING-WAVE

We return to the compression of the traveling-wave to illustrate two regimes of coherent behavior that arise when the latent is an advecting wave but the readout only observes a subset of coordinates.

Consider the long-range copy task, a benchmark designed to test long-range information retention (Graves et al., 2014; Arjovsky et al., 2016; Keller et al., 2024). The task consists of an input sequence of $N$ random scalar integers in $\{1, ..., 9\}$, followed by $T + N$ count of 0's. The target for this task is a sequence of the same length of all 0's except the last $N$ elements that are set to the initial sequence. The duration is $T + N$ and the resolved space is $\hat{y} \in \mathbb{R}^1$.

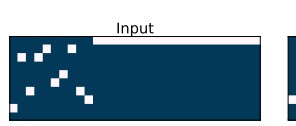 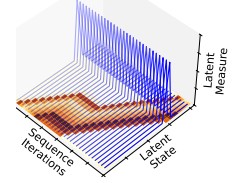 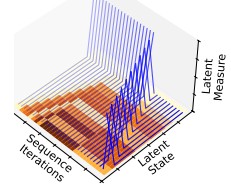

(a) Copy Task (One-Hot Visualization)   (b) Invariant-trivialization regime   (c) Projection-induced regime

Figure 1: (a) The copy task. (b) Invariant trivialization. The readout $P$ is fixed (blue), while the latent wave (orange) transports symbols to the readout location. (c) Projection-induced coherence. With constrained latent width, the latent saturates; a time-varying readout $\{P_k\}$ sweeps the compressed latent to reconstruct the sequence at the output.

Let $f_k \in \{0, \ldots, 9\}$ denote the input at time $k$. Let $S$ be a strictly subdiagonal shift, we introduce the softmax $\sigma$ and parameter $\theta$ to transition between $S$ and a parameterized matrix $W$

$$\boldsymbol{u}_{k+1} = \sigma(\theta)\boldsymbol{S}\boldsymbol{u}_k + (1 - \sigma(\theta))\boldsymbol{W}\boldsymbol{u}_k + \boldsymbol{e}_0 f_k, \quad \hat{\boldsymbol{y}}_k = \boldsymbol{P}_k \boldsymbol{u}_k,$$

where $\boldsymbol{e}_0$ injects the input at the left boundary, and $\boldsymbol{P}_k$ restricts the resolved coordinates/readout. When the latent has width $d$ and the task duration is $T + N$, two qualitatively distinct regimes appear as illustrated in Figure 1. In Figure 1(b) recall is carried entirely by the traveling wave with a fixed readout, and in Figure 1(c) the latent compresses and a time-varying readout sweeps over it. Both are examples of what we will refer to as coherent dynamics.

**Coherent dynamics** We call the dynamics *coherent* when either the latent wave or the readout subspace remains simple and persistent over extended intervals–e.g., a steady traveling front in the latent or a slowly varying/constant readout that consistently extracts stored content.

**Invariant trivialization (time-independent readout)** If the latent is wide enough to carry the entire history ($d \geq T + N + 1$), we can choose a time-independent projector ($\forall k, \boldsymbol{P} = \boldsymbol{P}_k$) that simply selects the coordinate where the advected information arrives at the end of the delay window (e.g., index $T + N + 1$). In this case $\hat{\boldsymbol{y}}_k = f_{k-(T+N+1)}$ exactly and all time-variation needed for recall is supplied by the latent advection; the readout is stationary. We refer to this as *invariant trivialization* because, after the (co-moving) lift that centers the wave, the projection subspace does not change over time. This is illustrated in Figure 1b where the blue readout remains flat while the orange latent front carries the symbols downstream.

**Projection-induced emergence (time-dependent readout)** When the latent is narrow ($d \leq T + N + 1$), the wave cannot store the entire history at distinct spatial positions. The learned evolution operator may deviate from the dynamics of the lift operator so that after the inputs cease (at step $N$), the latent often settles into a compact, stable pattern that retains only a compressed trace of the sequence. Successful recall then requires the projection itself to evolve in time-a family $\{P_k\}$ that sweeps across, or re-weights, the compressed latent so that the appropriate symbol is routed to the output at each step. Coherence therefore emerges from the projection dynamics: even though the latent has become stationary or a slow-moving manifold, the time-varying readout produces a systematic, wave-like replay at the output. This is illustrated in Figure 1c, and we term this projection-induced emergence.

These two coherent regimes in the simple copy serve as our motivating example. We formalize them using time-dependent projection operators and an intrinsic generalized Langevin equation.

# 3 MORI-ZWANZIG FORMALISM FOR COHERENCE AND EMERGENCE

We first formalize the two assumptions made, then our intrinsic GLE, and finally, coherence.

## 3.1 AN INTRINSIC TIME-DEPENDENT GENERALIZED LANGEVIN EQUATION

We model a fixed resolved subspace of information $\mathcal{V}_*$ which receives time-varying input from an unresolved space $g(t) \in \mathcal{W}_t$, and the aim is to output information $h(t) \in \mathcal{V}_t \subset \mathcal{V}_*$ from a time-varying subspace of the resolved space. That is, we assume $\mathcal{V}_*$ covers the entire family of incoming information $\{\mathcal{W}_t\}$, while the outputs evolve in the (possibly smaller) time-dependent subspaces $\{\mathcal{V}_t\}$. We formalize our lifting assumption as follows.

**Assumption 3.1.** *(Encoding Time-Dimension Tradeoff) The embedding of a generic input $g_t \in \mathcal{W}_t$ is a transport map $T_t : \mathcal{W}_t \to \mathcal{V}_*$ to a time-invariant subspace.*

For example, a traveling-wave is a co-moving shift, i.e., a linear lifting operator that increases dimensionality and centers moving patterns in the latent state. Kuramoto models are non-linear lifts that trade time-dependence in the signal for dimensions in phase coordinates.

We will learn a family of projection operators $\{P_{\mu_t}\}$ by parameterizing their measures $\{\mu_t\}$ over time. Consider a family of measures $\{\mu_t\}_{t \in [0,T]}$ with $\mu_t \ll \mu_*$ for all $t$, i.e. $\mu_t$ is absolutely continuous with respect to $\mu_*$. Our time-dependent projection operators are defined by

$$P_{\mu_t} : \mathcal{V} \to \mathcal{V}_t, \qquad P_{\mu_t} g = \mathbb{E}_{\mu_t}[g \mid \mathcal{G}], \quad \mathbb{E}_{\mu_t}[f \mid \mathcal{G}] = \frac{\mathbb{E}_{\mu_*}[\rho_t f \mid \mathcal{G}]}{\mathbb{E}_{\mu_*}[\rho_t \mid \mathcal{G}]},$$

which is the conditional-expectation onto the $\sigma$-algebra $\mathcal{G}$ of the fixed resolved space but with weights $\mu_t$. Note that $\rho_t = \frac{d\mu_t}{d\mu_{d0}}$ is the Radon-Nikodym derivative further discussed in Appendix C. We formalize our assumption of the derivative of $P_{\mu_t}$ as follows.

**Assumption 3.2.** *(Differentiability of $P_{\mu_t}$) Suppose the time-dependent conditional expectation operator $P_{\mu_t} : L^2(\mu_*) \to L^2(\mu_t)$ is Fréchet-differentiable with derivative $\dot{P}_{\mu_t}$.*

## 3.2 INTRINSIC TIME-DEPENDENT GLE

Here we present our GLE. Appendix C provides a formal derivation, with proofs in Appendix D.

**Proposition 3.1.** *(Intrinsic Time-Dependent GLE) Let $g(t)$ evolve under the operator $\mathcal{L}$ on a fixed Hilbert space $\mathcal{H} = L^2(\mathcal{M}, \mathcal{F}, \mu_*)$. Let $P_{\mu_*} : \mathcal{H} \to \mathcal{V} \subset \mathcal{H}$ be an orthogonal projection onto $\mathcal{V} = L^2(\mathcal{M}, \mathcal{G}, \mu_*)$ with $\mathcal{G} \subset \mathcal{F}$. For a family of $C^1$ measures $\{\mu_t\}_{t \in [0,T]}$ let $P_{\mu_t} : \mathcal{V} \to \mathcal{V}_t$ be the corresponding family of projections defining a Hilbert bundle $\{\mathcal{V}_t\}_{t \in [0,t]}$ with $\mathcal{V}_t = L^2(\mathcal{M}, \mathcal{G}, \mu_t)$. The evolution of the resolved variable $P_{\mu_t} g(t)$ satisfies the following GLE*

$$\frac{d}{dt}\left(P_{\mu_t} P_{\mu^*} g(t)\right) = P_{\mu_t} \dot{P}_{\mu_t} Q_{\mu_t} P_{\mu_*} g(t) + P_{\mu_t} \mathcal{L} P_{\mu_*} g(t) \tag{3}$$

$$+ \int_0^t P_{\mu_t} P_{\mu^*} \mathcal{L} e^{(t-s)Q_{\mu_*}\mathcal{L}} P_{\mu_*} g(s) ds + P_{\mu_t} \mathcal{L} e^{tQ_{\mu_*}\mathcal{L}} Q_{\mu^*} g(0).$$

The additional term $P_{\mu_t} \dot{P}_{\mu_t} Q_{\mu_t} P_{\mu_*} g(t)$ captures the instantaneous drift of the resolved state caused by the time-dependent rotation of the projection subspace, i.e., the transfer of latent information. This additional term is similar to the FFE-MZ. However, our approach does not result in a two-time memory kernel, and our drift depends only the dynamics of intrinsic subspaces $P_{\mu_t}$ and $Q_{\mu_t}$.

## 3.3 COHERENCE AND EMERGENCE

We now use the kinematic drift term from Proposition 3.1 to characterize how time-dependent projections give rise to coherent memory dynamics.

**Invariant trivialization** Our projections $P_{\mu_t}$ live on a family of $L^2(\mu_t)$ spaces whose inner products change with time. A trivialization is a way of re-expressing every $L^2(\mu_t)$ inside a single reference space $L^2(\mu_0)$. In general, as $t$ varies this change-of-measure will warp basic functions and

cause the projection subspace to rotate. Using the Radon-Nikodym densities (Appendix C), we obtain a spatially uniform change-of-measure. The trivialization becomes a global rescaling that does not distort the directions in $L^2$. In this case, there exists a basis that is preserved for all $t$, where the projection is time-invariant. We call this situation an *invariant trivialization*, and it corresponds to the lift-driven coherence regime from the copy example. We formalize this as follows.

**Proposition 3.2.** *(Coherence Under Invariant Trivialization) If the densities $\rho_t(x)$ are spatially constant, $\rho_t(x) = \alpha(t)$, then the family of subspaces $\{\mathcal{V}_t\}$ is unitarily equivalent to the fixed subspace $\mathcal{V}_0$. Then $\{P_{\mu_t}\}$ is coherent under the invariant trivialization $T_t$ where $T_t P_{\mu_t} T_t^{-1} = P_{\mu_t} = P_{\mu_0}$.*

In this case, the drift term will vanish, providing an **explainable and controllable mechanism.**

**Corollary 3.1.** *(Vanishing Drift Under an Invariant Trivialization) Suppose the Radon-Nikodym densities satisfy $\rho_t(x) = \alpha(t)$, and $\alpha > 0$ independent of $x$. Then $P_{\mu_t} = P_{\mu_0}$, hence $\dot{P}_{\mu_t} = 0$.*

**Projection-induced coherence**   In the intrinsic GLE of Proposition 3.1, the only place where time-dependence of the projection enters is through the kinematic drift term $P_{\mu_t} \dot{P}_{\mu_t} Q_{\mu_t}$. Intuitively, once we compress the latent dynamics onto an r-dimensional resolved subspace $\mathcal{V}_t$, any extra motion coming from $\dot{P}_{\mu_t}$ must be funneled through a small set of directions inside $\mathcal{V}_t$. We refer to this situation—where nonzero $D_t$ organizes the resolved dynamics along a few stable directions in $\mathcal{V}_t$—as projection-induced coherence. We make this statement precise in the following proposition.

**Proposition 3.3.** *(Low-rank drift under latent compression) Let $\mathcal{V}_t$ be the resolved subspace at time $t$, with dimension $\mathcal{V}_t = r$, and define the kinematic drift*

$$D_t := P_{\mu_t} \dot{P}_{\mu_t} Q_{\mu_t} : \mathcal{H} \to \mathcal{V}_t.$$

*Then $\mathrm{rank}(D_t) \leq r$. In particular, all additional drift induced by the time-dependence of the projection is confined to at most $r$ independent directions in $\mathcal{V}_t$.*

### 3.4   A NEURAL WAVE FIELD ARCHITECTURE

We instantiate the intrinsic GLE in a Neural Wave Field architecture that factorizes the latent evolution into a fixed lift and boundary update, an MZ-driven latent dynamics module, and a projection module illustrated in Figure 2. At each time step $t$, the input $\boldsymbol{x}_t$ is embedded as a *ghost boundary* into a 1-D latent field $\boldsymbol{h}_t \in \mathbb{R}^n$. A fixed shift operator $\boldsymbol{S}$ implements the traveling-wave lift (co-moving frame), while a small gating network latent update mixes three simple behaviors, identity, pure shift and direct boundary injection. This mixture controls how much new input overwrites or augments the boundary versus how much of the existing wave is transported downstream.

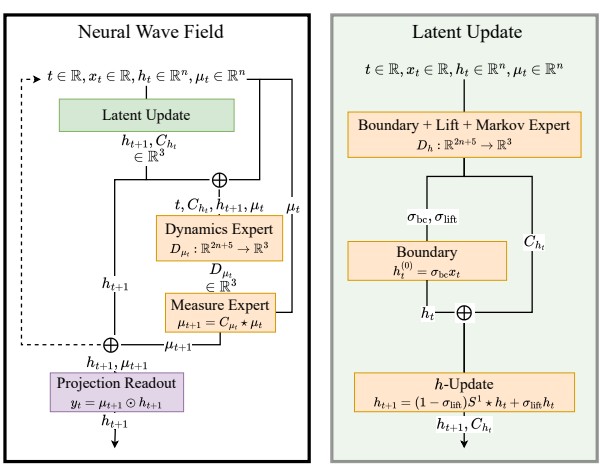

Figure 2: A diagram of the Neural Wave Field architecture.

The MZ driftand the projection dynamics are then modeled by two additional conditional experts. A dynamics expert $D_{\boldsymbol{h}_t}$ takes as context the current latent state, the boundary state, and the lift outptut, and returns coefficients that define a local convolution kernel $C_{\boldsymbol{h}}(\boldsymbol{h}_t, \boldsymbol{x}_t, t)$. This kernel parameterizes the instantaneous drift term $P_{\mu_t} \mathcal{L} P_{\mu_*}$ in the intrinsic GLE. In parallel, a measure expert $D_{\mu_t}$ operates on the current measure $\mu_t$, the latent features, and time to produce a normalized convolution kernel $C_{\mu_t}(\boldsymbol{h}_t, \mu_t, t)$ that updates the measure $\mu_{t+1}$ and thus the projection $P_{\mu_t}$. This yields a learned, context-dependent kinematic drift $C_{\mu_t}$. Under invariant trivialization, $C_{\mu_t}$ vanishes, while under projection-induced coherence it remains nonzero but low-rank. The final output is the element-wise product $y_t = \boldsymbol{h}_t \odot \mu_t$, so that the decoding is explicitly tied to the current projection.

## 4 EXPERIMENTAL RESULTS

To empirically evaluate our theoretical framework, we test our architectures ability to learn coherent dynamics for traveling-wave and non-linear oscillatory models. These results further support Proposition 3.1 across a range of task including long-range benchmarks and real-world EEGs. We find that the derived GFDR provides enhanced the robustness of the coherence across all tasks further supporting the use of MZ formalism. Furthermore, we demonstrate how these emergent behaviors can help characterize memory encoding, retention and retrieval similar to biological neural networks.

For a comparison on long-range benchmarks, we consider WaveRNN (Keller et al., 2024), Mamba (Gu & Dao, 2024), Alibi (Press et al., 2021), NoPe (Kazemnejad et al., 2023), and RoPe (Su et al., 2024). For a comparison on real-world data, we consider several baseline models including ShallowFBCSPNet Ang et al. (2008), Deep4 Schirrmeister et al. (2017), EEGNet Lawhern et al. (2018) and TIDNet Kostas & Rudzicz (2020). Additional details including hyperparameters and optimization procedures and can be found in Appendix F.

### 4.1 COPY TASK

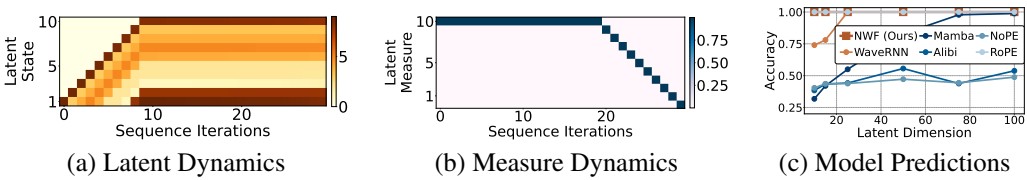

(a) Latent Dynamics      (b) Measure Dynamics      (c) Model Predictions

Figure 3: Explainability of the Neural Wave Field equipped with a traveling-wave lift operator in (a) the latent state (b) the measure and (c) a model comparison.

For comparison, we evaluate the accuracy of each architecture as the size of the latent stat is reduced. In particular, we systematically constrain the latent dimension from 100 to 10, the minimal size needed to represent the information in the long-range copy task for $T = 10$. This forces each model to rely on its latent dynamics rather than excess dimensionality, and allows us to test whether the core mechanism can efficiently encode and preserve information. In Figure 1(a), we see that our architecture maintains high accuracy even at the minimal latent dimension, where the information content fully saturates the latent state.

### 4.2 SELECTIVE COPY TASK

The selective copy task (Jing et al., 2019; Gu & Dao, 2024) modifies the copy task by randomizing the spacing of the $N$ tokens over the first $N + T$ inputs. The target is the same as the copy task. Due to this randomization, it requires more data-dependent reasoning to solve the task. From the FFE–MZ perspective, the task highlights how the lifting operation is tied to the time-dependent projections. Specifically, when the projection operator evolves in time, the lifting operator is static.

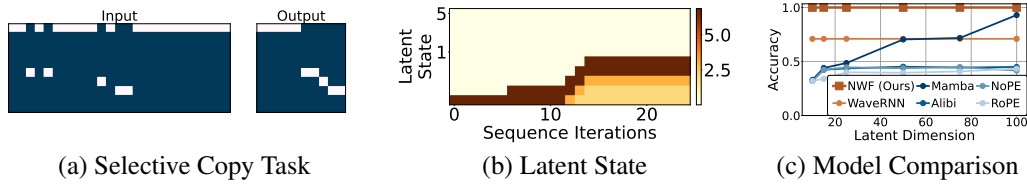

(a) Selective Copy Task      (b) Latent State      (c) Model Comparison

Figure 4: Results for the selective copy task (a) the latent state (b) the model predictions and (c) a memory capacity comparison against baselines.

In Figure 4 we illustrate the (a) latent state (b) model predictions and (c) a comparison of architectures as the memory capacity is reduced. As a result, of the tie between the lifting operator and the measure dynamics, we observe that the latent state encodes only the relevant information. Here it achieves a minimal yet sufficient representation in the latent state. As a result, it not only preserves accuracy but also outperforms all baseline architectures under identical constraints.

## 4.3 EEG DATASET

Using the braindecode library, we further benchmark our method on two neuroscience datasets from the BNCI IV competition 1) The BNCI IV-2a dataset, which contains EEG recordings from 9 subjects performing four motor imagery tasks: left hand, right hand, foot, and tongue movements. Each subject completed two sessions across different days, with 288 trials per session (12 per class per run, 6 runs per session). 2) The BNCI IV-4 dataset, which contains recordings and simultaneous finger-flexion measurements from three epilepsy patients at Harborview Hospital, Seattle. Each subject wore a subdural platinum-electrode grid (62, 48, and 64 channels for Subjects 1–3) sampled at 1000 Hz (0.15–200 Hz band-pass) and referenced to a common average; finger movements of all five digits were captured via a 5-sensor data glove (25 Hz, up-sampled to 1 kHz). This benchmark allows us to assess whether the traveling-wave inductive bias meaningfully improves decoding under practical EEG conditions.

To test whether latent propagating dynamics can serve as an effective inductive bias for EEG/ECoG decoding, we insert our Neural Wave Field module at the front of the network, directly operating on raw EEG/ECoG signals. This module compresses the raw sequence into a dynamic latent state by simulating learned traveling waves in feature space using gated, memory-aware updates derived from the Mori–Zwanzig formalism. The output is a sequence representation that is then passed into a standard CNN-based classification pipeline, similar to ShallowFBCSPNet. In this way, we can assess if the latent traveling wave representation enhances the expressivity of the models.

By placing the NeuralField before conventional spatial-temporal filtering, we evaluate whether traveling-wave dynamics can serve as an effective neural preprocessor, enhancing downstream performance. This setting allows us to test the expressiveness and utility of our proposed inductive bias in a realistic, cue-based EEG classification task.

Table 1 presents the accuracy on the BNCI IV-2a dataset and the Pearson r-score on the BNCI IV-4 dataset. The Neural Wave Field is the second best performer with a single channel latent state size of 30, which maintains a compressed traveling wave representation of the full 22 channel

| Model | BNCI IV-2a (Accuracy ↑) | BNCI IV-4 ($r$-value ↑) |
|---|---|---|
| ShallowFBCSPNet | 72.9 | 0.311 |
| Deep4 | 56.25 | **0.653** |
| EEGNet | **77.08** | 0.354 |
| TIDNet | 40.97 | 0.356 |
| NWF (Ours) | 74.31 | 0.375 |

Table 1: Accuracy on the BNCI IV-2a dataset and Pearson's $r$ on the BNCI IV-4 dataset. The Neural Wave Field including a traveling wave lifting operator ranks as the second-best performer.

input. Again the Neural Wave Field is the second best performer with a single channel latent state size of 20, obtaining a compressed traveling wave representation of the full 62 channel input. Moreover, it showed strong improvement over the direct baseline ShallowFBCSPNet. It also suggest the potential to include alternative lifting operators that align better with the underlying dynamics.

## 5 CONCLUSION

We introduced an intrinsic time-dependent framework for the Mori-Zwanzig formalism and used it to derive a structured model of latent memory dynamics. In particular we observed how a lifting operator and the latent drift were coupled. Building on this, we proposed the Neural Wave Field architecture, which utilizes traveling wave lifting operations to learn both drift and memory closure end-to-end. Empirically, we validated our theoretical observations about the expressivity of the architecture, and showed that it reliably discovers coherent memory structures, achieves minimal latent representations and outperforms baselines on long-range sequence tasks.

**Limitations and Future Work**  While our Neural Wave Field provides a clear proof of concept, it is only one instantiation of a much richer framework to be explored in future works. In particular, our preliminary insights into EEG and ECoG datasets warrant further exploration of oscillatory models as lifting mechanisms. We made two assumptions regarding continuity and support of the measure $\mu_t$ in our framework. Empirically the first assumption stabilizes training as shown in the copy task of Section 4. The second assumption on the differentiability of $\mu_t$ may not always be assumed, e.g. for the ordered recall task a variant of the copy task in which numbers are recalled in order. A framework that handles discontinuous $\mu_t$ is non-trivial and would provide additional insights into higher level cognitive capabilities and we leave this to future work.

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

# A  RELATED WORKS

Data-driven MZ (Chorin et al., 2002; Lin et al., 2021) and time-delay embeddings (Brunton et al., 2017; Woodward et al., 2025; Zhu et al., 2021; Ostrow et al., 2024) fix a static projection (e.g., EDMD modes or a stack of delays) to learn stationary memory kernels. Deep learning extensions of these approaches use sequential inputs to learn memory kernels (Lin et al., 2023) in autoencoders (Gupta et al., 2024) and neural operators (Buitrago et al., 2025). Time-dependent MZ formalism has been used to characterized deep learning (Venturi & Li, 2023).

**Structured models.** Neural oscillators (Lanthaler et al., 2023; Rusch & Mishra, 2021; Keller & Welling, 2023), traveling-wave networks (Keller et al., 2024; Liboni et al., 2025; Keller, 2025), attractor embeddings (Ságodi et al., 2024), reservoir computing (Jaeger, 2007; Lukoševičius & Jaeger, 2009) and neural delay-difference equations (Zhu et al., 2021) bake in known dynamical motifs to encode memory explicitly. WaveRNN (Keller et al., 2024) structures its latent updates as linear advection, yielding transparent memory dynamics as traveling waves.

**Models.** GRUs, LSTMs, residual and deep equilibrium models are well established recurrent and feed-forward NNs with augmented memory-mechanisms. Continuous-time models such as neural ODEs (Chen et al., 2018) and ODE-RNNs (Rubanova et al., 2019) encode history through flows in state space. Recent state-of-the-art performance has been achieved by structured state-space-models (SSMs) (Gu et al., 2022; Fu et al., 2023; Gu & Dao, 2024), notably the oscillatory SSM–LinOSS (Rusch & Rus, 2025).

**Global context embeddings.** Transformers (Press et al., 2021; Su et al., 2024; Gumma et al., 2024) use full self-attention for global sequence dependencies. Recent work investigating the performance of positional encodings (Jelassi et al., 2024) has demonstrated that various positional encoding strategies (Press et al., 2021; Kazemnejad et al., 2023; Su et al., 2024) outperform SSMs on copying tasks.

**Memory Neural Operator (MemNO)** MemNO (Buitrago et al., 2025) interleaves a memory operator (sequential model) into the layer updates of a neural operator, in order to capture memory effects in a GLE inspired manner. The goal of the memory operator is to re-introduce projected variables, and is theoretically motivated by a theorem demonstrating the divergence of solutions with and without memory for a second-order elliptic PDE. The approach is empirically validated by testing super resolution capacity of architectures, i.e. reducing the input resolution and maintaining the output resolution during the training of an encoder-decoder framework. A further ablation study is performed on the window size of the memory operator, where the performance improves as the window size increases, i.e. in a time-delay embedding fashion.

At a high level, both works aim to resolve a memory closure using sequential linear layers (S4 in the case of MemNO). MemNO uses a multi-layer FNO as an embedding and read out of the latent state, whereas NWF uses linear layers based on projection operators. Additionally, NWF directly induces wave like phenomena into the latent state, and studies the rise of phenomena like coherence and emergence. An interesting future direction would be to characterize MemNO's memory operator using the theory developed here-in.

**Time-dependent GLE** This time-dependent relevant ensemble $\rho(t)$ has been extended to a bundle of trajectories, i.e. measurements for a distribution of moving points in the phase space (Meyer et al., 2017). The resolved subspace is changing in time and the two-time memory kernel appears again. The time-dependent projection operator is an average over all possible trajectories.

A discrete analogue of the time-dependent GLE has been proposed in the context of deep residual neural networks (Venturi & Li, 2023). In this formulation, each layer $n$ is associated with a projection operator $P_n$ and the hidden state evolves with a Markov term, two-time memory kernel and layer-wise fluctuating force. Although their streaming term does not explicitly include a kinematic component, it implicitly accounts for the evolution of the projection subspace across layers through the residual propagator. While (Venturi & Li, 2023) notes that MZ formalism can be used to reduce the total number of degrees of freedom in the neural network, in practice their approach does not provide a mechanism by which they may go about reducing the number of variables.

# B  MOTIVATING EXAMPLE

## B.1  BOUNDARY CONDITIONS

**Corollary B.1.** *(Toroidal Latent Manifold) Suppose we constrain each latent coorindate $h_i(t)$ to live on a circle of period $L_i$ and we enforce that both the learned drift and memory-kernel parameters depend on $h$ only through these periodic coordinates. Then the entire latent trajectory $h(t)$ evolves on the $m$-dimensional torus $\mathbb{T}^m$. As a result, the network can only represent–and learn–functions defined on this compact, boundary-free manifold.*

*Proof.* By the assumption of periodicity then each of the MZ terms descent to well-defined maps on the quotient $\mathbb{R}^m/(L_1\mathbb{Z} \times \ldots \times L_m\mathbb{Z})$, and the initial condition $h(0) \in S^1_{L_1} \times \ldots \times S^m_{L_m}$ uniquely determines a solution $h(t)$ that never leaves the torus.

Therefore any decoder $F : \mathbb{R}^m \to Y$ must descent to a well-defined map $\hat{F} : \mathbb{T}^m \to Y$, i.e., those maps that are periodic in each coordinate. □

# C  MORI-ZWANZIG FORMALISM

In this section, we present the preliminary background. We treat the latent state of the neural network as observations of an underlying dynamical system. The near-equilibrium MZ formalism (NE-MZ) describes the evolution of a time-invariant subset of observations. Time invariance may be overly restrictive for the latent states of a neural network, in which case we employ time-dependent operator formalism for far-from-equilibrium systems (FFE-MZ). Finally, we recall the important distinction of MZ-type memory, the generalized fluctuation-dissipation relation (GFDR).

## C.1  PRELIMINARIES

Suppose the underlying system evolves dynamically on a smooth manifold $\mathcal{M} \subset \mathbb{R}^n$, called the phase-state, described by the following (ergodic and possibly nonlinear) autonomous ODE

$$\frac{d\Phi(t)}{dt} = S\big(\Phi(t)\big), \quad \Phi(0) = x_0, \tag{4}$$

where $S : \mathcal{M} \to \mathbb{R}^n$ is $C^1$. By the Picard-Lindelöf (Coddington, 1955) theorem, Equation 4 admits a unique solution $\Phi_t(x_0) = \Phi(t)$ for all $t$ in $\mathbb{T} \subseteq \mathbb{R}$, inducing the flow $\Phi_t : \mathcal{M} \to \mathcal{M}$.

Let the collection $(\mathcal{M}, \mathcal{F}, \mu)$ be the phase-state manifold $\mathcal{M}$ equipped with a $\sigma$-algebra $\mathcal{F}$ and a finite, flow-invariant probability measure $\mu$. A system *observation* $g : \mathcal{M} \to \mathbb{R}$ is a real-valued square-integrable function, i.e. $g \in \mathcal{H} := L^2(\mathcal{M}, \mu)$ where $\mathcal{H}$ is a *separable* Hilbert space.

**Definition C.1.** *(Liouville Operator) The Liouville operator $\mathcal{L} : \mathcal{H} \to \mathcal{H}$ describes the infinitesimal evolution of an observable $g \in \mathcal{H}$ along the flow $\Phi_t$. In general we will take it to be $\frac{d}{dt}g(t) = \mathcal{L}g(t)$.*

Remarkably, the evolution of the observations can be expressed in terms of linear operators on $\mathcal{H}$, despite the underlying system being possibly nonlinear and mildly complex (ergodic). However, this is a linear operator that acts on the space of all observables, which may be infinite dimensional.

## C.2  NEAR-EQUILIBRIUM MORI-ZWANZIG FORMALISM (NE-MZ)

Using the separability of $\mathcal{H}$, the space of observations can be separated into a set of *resolved* observables and complementary *unresolved* observables. In particular, for any closed subspace $\mathcal{V} \subset \mathcal{H}$ there is a decomposition $\mathcal{H} = \mathcal{V} \oplus \mathcal{V}^\top$ realized by the unique orthogonal projection

$$P : \mathcal{H} \to \mathcal{V}, \quad Q = I - P : \mathcal{H} \to \mathcal{V}^\top \quad (P^2 = P, \, Q^2 = Q, \, P = P^*, \, Q = Q^*, \, PQ = 0).$$

These projections can be linear operators (Mori, 1965a), or as we adopt, (non)-linear operators (Zwanzig, 2001) realized as conditional expectations $P = \mathbb{E}\left[\,\cdot\,|\,\mathcal{G}\,\right]$ on the sub-$\sigma$-algebra $\mathcal{G} \subset \mathcal{F}$. In NE-ZM formalism the subset of observables–and therefore the projection $P$–is *time-invariant*.

**The generalized Langevin equation.** NE-MZ formalism describes the exact evolution of a time-invariant subset of observables by decomposing $\mathcal{H}$ into *resolved* $\hat{g} \in \mathcal{V}$ and *unresolved* $\tilde{g} \in \mathcal{V}^\top$ observables. The result is the generalized Langevin equation (GLE)

$$\frac{\partial}{\partial t}\hat{g}(t) = \underbrace{P\mathcal{L}\,\hat{g}(t)}_{\text{Markov}} + \underbrace{\int_0^t P\mathcal{L}\,e^{(t-s)Q\mathcal{L}}\,Q\mathcal{L}\,\hat{g}(s)\,\mathrm{d}s}_{\text{Memory}} + \underbrace{P\mathcal{L}e^{tQ\mathcal{L}}\,Q\,g(0).}_{\text{Fluctuating Force}} \tag{2}$$

Equation 1 consists of three distinct terms (underscored). The Markov term represents the instantaneous drift from the resolved dynamics. The Memory term re-introduces the influence of dynamics previously *forgotten*, i.e. prior resolved information that has been projected into the unresolved subspace. The Fluctuating Force term[2] captures the residual influence of the unresolved initial state.

## C.3 FAR-FROM-EQUILIBRIUM MORI-ZWANZIG FORMALISM (FFE-MZ)

For a neural network architecture, it may not be possible–and potentially unreasonable–to ascribe to each element of its latent state a static representation. Our approach models the latent state by using FFE-MZ (Grabert, 2006) which allows $P(t)$ to evolve.

The resulting GLE is given by

$$\frac{\partial}{\partial t}\hat{g}(t) = P(t)\mathcal{L}\hat{g}(t) + \dot{P}(t)g(t) + \int_0^t P(t)\mathcal{L}G(t,s)Q(s)\mathcal{L}\hat{g}(s)\mathrm{d}s + P(t)\mathcal{L}G(t,0)Q(0)\,g(0). \tag{5}$$

The two-time memory kernel $G(t,s) = \mathcal{T}_- \exp\!\left(\int_s^t Q(u)\,\mathcal{L}\,\mathrm{d}u\right)$ is the negatively time-ordered exponential operator that captures the extrinsic influence from the evolution of the subspaces. The Kinematic term $\dot{P}(t)g(t)$ captures the intrinsic evolution of the resolved subspace (Meyer et al., 2017).

Critically, the time-dependent projection operator acts as moving frame of reference that is tied to the relevant ensemble. The source of the time dependence is *extrinsic* to the resolved observables i.e. it is driven. As a result, this non-stationarity cannot be removed by a simple change of coordinates.

## C.4 GENERALIZED FLUCTUATION DISSIPATION RELATION (GFDR)

We now observe the critical distinction between MZ memory and auto-regressive or time-delay mechanisms, that MZ assumes an underlying principle of detailed balance. The principle of detailed balance states that at equilibrium, each process is in equilibrium with its reverse process. For NE-MZ this is formalized via the fluctuation-dissipation theorem (Callen & Welton, 1951) directly. The *generalized* fluctuation-dissipation relation (GFDR) is the extension to FFE-MZ (Meyer et al., 2019)

$$K(t,s) = \langle F(t|s), F(s)\rangle C(s)^{-1} \tag{6}$$

$$K(t,s) = P(t)\mathcal{L}G(t,s)Q(s)\mathcal{L}, \ C(s) = \langle \hat{g}(s), \hat{g}(s)\rangle, \ F(s) = Q(s)\mathcal{L}\hat{g}(s), \ F(t|s) = G(t,s)F(s)$$

which relates the memory kernel $K(t,s)$ to the level of noise $\langle F(t|s), F(s)\rangle$ relative to the covariance of the resolved observable $C$. Instead of treating noise in the black-box model as a limitation of explainability, MZ formalism allows us to model noise predictably from the memory kernel itself.

For more details on the distinction of architectures, we refer the reader to (Lin et al., 2023).

## C.5 INFORMAL DERIVATIONS

**The derivation of the GLE.** The instantaneous evolution of $g$ is given by

$$\frac{d}{dt}e^{t\mathcal{L}}g(0) = \mathcal{L}e^{t\mathcal{L}}g(0),$$

which can be decomposed into its two projected dynamics yeilding two coupled equations

$$\frac{d}{dt}Pe^{t\mathcal{L}}g(0) = P\mathcal{L}Pe^{t\mathcal{L}}g(0) + P\mathcal{L}Qe^{t\mathcal{L}}g(0),$$

$$\frac{d}{dt}Qe^{t\mathcal{L}}g(0) = Q\mathcal{L}Qe^{t\mathcal{L}}g(0) + Q\mathcal{L}Pe^{t\mathcal{L}}g(0).$$

---

[2] The third term referred to by (Mori, 1965a) as a random force and by (Zwanzig, 2001) as a fluctuating force, is frequently called the noise term in data-driven and stochastic applications.

We rewrite the second equation for $v(t) = Qe^{t\mathcal{L}}g(0)$ where $A(t) = Qe^{t\mathcal{L}}g(0)$ and $F(t) = Q\mathcal{L}Pe^{t\mathcal{L}}g(0)$,

$$\frac{d}{dt}v(t) = A(t)v(t) + F(t).$$

The solution is given by Dyson's identity

$$v(t) = e^{tA}v(0) + \int_0^t e^{(t-s)A}F(s)ds.$$

Notice that $v(0) = Qg(0)$. Substituting for $v, A, F$, we have

$$Qe^{t\mathcal{L}}g(0) = e^{tQ\mathcal{L}}Qg(0) + \int_0^t e^{(t-s)Q\mathcal{L}}Pe^{s\mathcal{L}}g(0)ds = e^{tQ\mathcal{L}}g(0) + \int_0^t e^{(t-s)Q\mathcal{L}}Pg(s)ds.$$

The GLE results from substituting the prior result into the dynamics for $\frac{d}{dt}Pg(t)$

$$\frac{\partial}{\partial t}Pg(t) = P\mathcal{L}Pg(t) + \int_0^t P\mathcal{L}e^{(t-s)Q\mathcal{L}}Q\mathcal{L}Pg(s)\,\mathrm{d}s + P\mathcal{L}e^{tQ\mathcal{L}}Qg(0).$$

**The connection to Koopman operator theory.** The Koopman operator $\mathcal{K}^t : \mathcal{H} \to \mathcal{H}$ is a bounded linear operator that evolves any observable $g \in \mathcal{H}$ along the flow $T \subset \mathbb{R}$ on the phase manifold

$$\mathcal{K}^t g(x_0) = g(T(x_0, t)).$$

Because $\mathcal{H}$ is infinite dimensional, in practice one often restricts attention to a finite resolved subspace $\mathcal{V} = \mathrm{Span}\{g^{(1)}, \ldots, g^{(r)}\} \subset \mathcal{H}$ with orthogonal complement $\mathcal{V}^{\top}$.

The evolution of $\hat{g} \in \mathcal{V}$ in this reduced subspace, with restricted evolution operator $\widehat{\mathcal{K}}$, accumulates an error term

$$\hat{g} \circ T = \widehat{\mathcal{K}}^t \hat{g} + r,$$
$$r \in \mathcal{V}^{\top}.$$

where $\hat{g} \in \mathcal{V}$. The residual $r$ is the closure problem, which is addressed via the Mori–Zwanzig formalism by projecting onto $\mathcal{V}$ while accounting for the influence of $\mathcal{V}^{\top}$.

# D   THEORETICAL DETAILS

In this section, we provide proofs of the corresponding propositions from Section 3.

## D.1   ASSUMPTIONS

For completeness, we restate our assumptions below. In addition, we will provide some more context to the significance of these assumptions.

**Assumption 3.2.** *(Differentiability of $P_{\mu_t}$) Suppose the time-dependent conditional expectation operator $P_{\mu_t} : L^2(\mu_*) \to L^2(\mu_t)$ is Fréchet-differentiable with derivative $\dot{P}_{\mu_t}$.*

This assumption is critical to ensuring that the GLE is well-posed. In practice it forces us to choose a feature-map basis whose dependence on $t$ makes $P_{\mu_t}$ a smooth function of time—only then can the model reliably learn the evolving dynamics.

**Assumption D.1.** *(Support Coverage Assumption) Let $\tilde{\mu} = \sum_{t=1}^T \mu_t$. We require $\mu_* \ll \tilde{\mu}$ or equivalently $\mathrm{supp}(\mu_*) \subseteq \bigcup_{t=1}^T \mathrm{supp}(\mu_t)$.*

Assumption D.1 ensures that every region with positive mass under $\mu_*$ is observed at some time $t$, so that all potential degrees of freedom in the reference measure are, in principle, observable. This condition ensures that the projected dynamics $P_{\mu_t}$ can act on the entire latent state: there are no hidden modes in $\mu_*$ that fall completely outside the supports of the training measures. Equivalently, it removes any degrees of freedom from the latent state, so that our GLE truly governs all of the relevant latent dynamics.

## D.2 Proofs

**Proposition 3.1.** *(Intrinsic Time-Dependent GLE) Let $g(t)$ evolve under the operator $\mathcal{L}$ on a fixed Hilbert space $\mathcal{H} = L^2(\mathcal{M}, \mathcal{F}, \mu_*)$. Let $P_{\mu_*} : \mathcal{H} \to \mathcal{V} \subset \mathcal{H}$ be an orthogonal projection onto $\mathcal{V} = L^2(\mathcal{M}, \mathcal{G}, \mu_*)$ with $\mathcal{G} \subset \mathcal{F}$. For a family of $C^1$ measures $\{\mu_t\}_{t\in[0,T]}$ let $P_{\mu_t} : \mathcal{V} \to \mathcal{V}_t$ be the corresponding family of projections defining a Hilbert bundle $\{\mathcal{V}_t\}_{t\in[0,t]}$ with $\mathcal{V}_t = L^2(\mathcal{M}, \mathcal{G}, \mu_t)$. The evolution of the resolved variable $P_{\mu_t} g(t)$ satisfies the following GLE*

$$\frac{d}{dt}\left(P_{\mu_t} P_{\mu_*} g(t)\right) = P_{\mu_t} \dot{P}_{\mu_t} Q_{\mu_t} P_{\mu_*} g(t) + P_{\mu_t} \mathcal{L} P_{\mu_*} g(t) \tag{3}$$

$$+ \int_0^t P_{\mu_t} P_{\mu_*} \mathcal{L} e^{(t-s)Q_{\mu_*}\mathcal{L}} P_{\mu_*} g(s) ds + P_{\mu_t} \mathcal{L} e^{tQ_{\mu_*}\mathcal{L}} Q_{\mu_*} g(0).$$

*Proof.* By Assumption 3.2 $P_{\mu_t}$ is differentiable, so that the GLE is given by chain rule as

$$\frac{d}{dt}\left(P_{\mu_t} P_{\mu_*} g(t)\right) = \dot{P}_{\mu_t} P_{\mu_*} g(t) + P_{\mu_t} P_{\mu_*} \frac{d}{dt} g(t) = \dot{P}_{\mu_t} P_{\mu_*} g(t) + P_{\mu_t} P_{\mu_*} \mathcal{L} g(t).$$

Let $\mathcal{H}$ and $\mathcal{V}$ be decomposed as $\mathcal{H} = \mathcal{V} \oplus \mathcal{V}^\top = \operatorname{ran}(P_{\mu_*}) \oplus \operatorname{ran}(Q_{\mu_*})$, and $\mathcal{V} = \operatorname{ran}(P_{\mu_t}) \oplus \operatorname{ran}(Q_{\mu_t})$ for all $t$. First, using the decomposition of $\mathcal{V}$, we rewrite

$$\dot{P}_{\mu_t} P_{\mu_*} g(t) = \dot{P}_{\mu_t} P_{\mu_t} P_{\mu_*} g(t) + \dot{P}_{\mu_t} Q_{\mu_t} P_{\mu_*} g(t)$$

using the identities in Section 2. Multiplying on the left by $P_{\mu_t}$ yields $P_{\mu_t} \dot{P}_{\mu_t} P_{\mu_t} = 0$, so $P_{\mu_t} \dot{P}_{\mu_t} \hat{g}(t) = P_{\mu_t} \dot{P}_{\mu_t} Q_{\mu_t} \hat{g}(t)$.

Inserting the fixed-time decomposition for $\mathcal{L}g(t)$, we see $P_{\mu_t} \mathcal{L} g(t) = P_{\mu_t} \mathcal{L}(P_{\mu_*} + Q_{\mu_*}) g(t)$ hence

$$\frac{d}{dt}\left(P_{\mu_t} P_{\mu_*} g(t)\right) = P_{\mu_t} \dot{P}_{\mu_t} P_{\mu_*} g(t) + P_{\mu_t} \mathcal{L} P_{\mu_*} g(t) + P_{\mu_t} \mathcal{L} Q_{\mu_*} g(t)$$

Finally, using Dyson's identity to solve for $v(t) = Q_{\mu_*} g(t)$ as in the standard MZ formalism, we find

$$\frac{d}{dt}\left(P_{\mu_t} P_{\mu_*} g(t)\right) = P_{\mu_t} \dot{P}_{\mu_t} Q_{\mu_t} P_{\mu_*} g(t) + P_{\mu_t} \mathcal{L} P_{\mu_*} g(t)$$

$$+ \int_0^t P_{\mu_t} P_{\mu_*} \mathcal{L} e^{(t-s)Q_{\mu_*}\mathcal{L}} P_{\mu_*} g(s) ds + P_{\mu_t} \mathcal{L} e^{tQ_{\mu_*}\mathcal{L}} Q_{\mu_*} g(0).$$

$\square$

**Corollary 3.1.** *(Vanishing Drift Under an Invariant Trivialization) Suppose the Radon-Nikodym densities satisfy $\rho_t(x) = \alpha(t)$, and $\alpha > 0$ independent of $x$. Then $P_{\mu_t} = P_{\mu_0}$, hence $\dot{P}_{\mu_t} = 0$.*

*Proof.* For any $g \in L^2(\mathcal{M}, \mu_t)$, $P_{\mu_t}$ is defined by the requirement

$$\int_G f d\mu_t = \int_G (P_{\mu_t} f) d\mu_t \qquad \text{for all measurable } G.$$

Since $\mu_t = \alpha(t)\mu_0$

$$\int_G f d\mu_t = \alpha(t) \int_G f d\mu_0, \qquad \int_G (P_{\mu_t} f) d\mu_t = \alpha(t) \int_G (P_{\mu_t} f) d\mu_0$$

Therefore

$$\int_G f d\mu_0 = \int_G (P_{\mu_t} f) d\mu_0 \qquad \text{for all measurable } G,$$

which by the uniqueness of the conditional-expectation operator in $L^2$ characterizes $P_{\mu_0}$. We thus conclude that $P_{\mu_t} = P_{\mu_0}$ for all $t$, and as a result the time-derivative vanishes, i.e., $\dot{P}_{\mu_t} = 0$. $\square$

**Note on trivialization and isometries.** Collectively, the family $\{\mathcal{V}_t\}_{t \in [0,T]}$ together with the projection map

$$\pi = \bigsqcup_t \mathcal{V}_t \to [0, T], \qquad \pi(v) = t$$

constitutes a Hilbert bundle over the interval $[0, T]$. In this bundle picture, fibers are the individual $\mathcal{V}_t$, a section is a time-indexed observable $g(t) \in \mathcal{V}_t$. Here we describe trivialization with respect to a fixed reference within the bundle, which is given by the Radon-Nikodym isometry

$$\mathcal{T}_t : \mathcal{V}_0 \to \mathcal{V}_t, \qquad \mathcal{T}_t(g) = \sqrt{\frac{d\mu_t}{d\mu_0}(x)} g(x) = \rho_t^{\frac{1}{2}} g.$$

# E    Intrinsic Generalized Fluctuation Dissipation Relation

**Assumption E.1.** *(Intrinsic Time-Dependent GFDR) For the fixed reference measure $\mu^*$ and projection $P_{\mu^*}$, the standard Mori–Zwanzig memory kernel and fluctuating force satisfy the generalized fluctuation–dissipation relation (5). In our intrinsic setting we enforce that the learned memory kernel $K_{int}$ and noise $F_{int}$ are compatible with this structure in the sense that*

$$K_{int}(t - s) = P_{\mu_t} K_{\mu^*}(t - s), \qquad F_{int}(t) = P_{\mu_t} F_{\mu^*}(t),$$

*where $K_{\mu^*}, F_{\mu^*}$ satisfy the GFDR under $\mu^*$.*

# F    Methodological Details

## F.1    Architectural Details

**Neural Wave Field**    The Neural Wave Field maintains two coupled latent state $h_t \in \mathbb{R}^n$ and $\mu_t \in \mathbb{R}^n$, which evolve under a Mori-Zwanzig inspired network and an accompanying measure-update expert. At each time step $t$ the raw input $x_t$ is first embedded into the feature space as a ghost boundary point. That is, it is available to be uptaken by the memory kernel provided the gating mechanism allows it.

For this reason, the MZ-NET $\sigma_{\text{mem}}$ and $\sigma_{\text{force}}$ are critical for determining the amount of long history information to retain, and the amount of new information to incorporate into the memory state. Whether the information is ultimately taken into the latent state is governed by $\sigma_{\text{closure}}$. These signals jointly determine a convolutional kernel $C_{h_t}$ and padded hidden state $\tilde{h}_t$ for updating $h_{t+1} = C_{h_t} \star \tilde{h}_t$.

A measure-dynamics expert network $D_\mu$ determines the update for the measure between two time periods. This module enforces that $\mu_t$ remains a valid probability density via softmax with a large temperature of 100.

Given our assumptions on the conditional-expectation projections of $P_{\mu_t}$, we train using the MSE loss across all tasks.

**WaveRNN**    The WaveRNN architecture is most similar to the Neural Wave Field in its construction of a latent state. There are two particular differences in the approaches. First, the WaveRNN utilizes periodic boundary conditions which are a limiting factor as described by Corollary B.1. Moreover, the architecture relies on a static decoder and encoder which forces the projection dynamics to be invariant. As a result, the architecture will be unable to achieve a minimal latent state representation. Furthermore, it will be prohibited from accurately learning the selective copy task.

**Mamba**    The Mamba architecture is a state-of-the-art structured state-space model. It has achieved particular success in modeling long-range tasks. It has done so by balancing long-range and short range updates to the latent state.

**Transformers**    The positional encoding-based (or replacement) transformers aim to use various methods to replace fixed positional encoding mechanisms with relative positional encoding mechanisms. These have shown strong results in memory tasks such as the copy task.

### F.2 ADDITIONAL EXPERIMENTS

#### F.2.1 CHAOTIC DYNAMICAL SYSTEMS

We evaluate how well our architecture can learn a highly non-periodic, chaotic manifold in accordance with Corollary B.1. For this reason, we compare against the WaveRNN baseline, which uses periodic boundary conditions in its latent state. We train both models to reconstruct the full phase-state from only its $x$-coordinate, using 300-step input sequences ($\Delta t = 0.01$), and a latent dimension of 3.

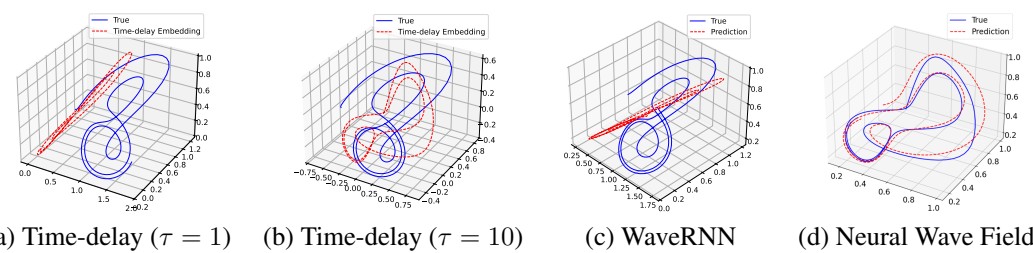

(a) Time-delay ($\tau = 1$)  (b) Time-delay ($\tau = 10$)  (c) WaveRNN  (d) Neural Wave Field

Figure 5: Time-delay and predicted trajectories of the Lorenz attractor using the time delays of $\tau = 1$ and 10, and WaveRNN and Neural Wave Field models. We observe that the WaveRNN performs comparably to the under resolved $\tau = 1$ time-delay embedding. In contrast, the Neural Wave Field achieves strong trajectory matching that degrades over time as errors slowly accumulate.

Figure 5 presents the time-delay and predicted latent trajectories of the Lorenz attractor using two classical delay embeddings ($\tau = 1$ and $\tau = 10$), as well as the learned embeddings form the WaveRNN and Neural Wave Field models. In our Neural Wave Field model, the latent trajectory forms smooth, closed loops that align with the true attractor and only gradually diverge as errors accumulate. Although this is slightly relaxed behavior from Proposition **??**, it is attributable to approximation errors in the memory kernel, the drift dynamics, and the dynamics of $\mu_t$. By contrast, the WaveRNN fits the dynamics into a toroidal manifold introducing distortion and misalignment, especially over long time horizons, coinciding with Corollary B.1.

### F.3 TASK DETAILS

**Lorenz Attractor**    We simulate the Lorenz system

$$\dot{x} = \sigma(y - x), \qquad \dot{y} = x(\rho - z) - y, \qquad \dot{z} = xy - \beta z$$

with standard parameters $(\sigma, \rho, \beta) = (10, 28, 8/3)$ using a fourth-order Runge-Kutta integrator at step size $\Delta t = 0.01$. At each time step only the $x$-coordinate is provided as input; the models must reconstruct the full state $(x_t, y_t, z_t)$.

For all experiments, we use a training batch size of 128 and test using a batch size of 32. All batches are generated randomly to obtain the trajectory of 300 time-steps. The loss is only computed on the last 280 time-steps. For all models we use the Adam optimizer with a learning rate of 0.001 for 1000 batches.

For our comparisons, we use the following configurations. For WaveRNN (Keller et al., 2024), we use one channel, an identity activation, and a hidden dimension of 20 to have a more direct comparison to our model. The loss is mean squared error (MSE).

**Copy**    For all experiments, we use a training batch size of 128 and test using a batch size of 50. All batches are generated randomly to obtain the sequence of 10 tokens to be memorized. We use $T = 20$, so the total sequence length is 30. The loss is only computed on the last 10 tokens; the intermediate outputs are not considered. That is, we only care about the model's ability to reproduce the sequence of 10 tokens at the final 10 timesteps. For all models we use the Adam optimizer with a learning rate of 0.001 for 1000 batches.

For our comparisons, we use the following configurations. For WaveRNN (Keller et al., 2024), we use one channel and an identity activation to have a more direct comparison to our model. The loss

is mean squared error (MSE). For Mamba and the transformer models, we use cross entropy loss, as they naturally output logits over the vocabulary size. We found that these models needed at least 2 layers to perform on the task, which we use in our experiments. For the transformers, we use a single attention head.

**Selective Copy**   By randomizing token positions and focusing evaluation solely on the terminal outputs, this task highlights each model's ability to selectively attend to and retain the correct information. Our architecture's time-dependent projection and delay-coordinate closure enable it to isolate the $N$ informative tokens with minimal overhead, even as memory capacity is constrained.

## F.4   ASSUMPTIONS NOTE

As a note on the practical implications of the assumptions made. When the size of the latent state is larger than the minimal representation but not large enough to trivialize the dynamics of the measure, then the additional degrees of freedom provide many non-unique and non-trivial solutions. In this case, we experience large standard deviations in the training loss between runs with differing initial conditions. In the case where memory is sufficiently large to trivialize the measure dynamics, the learning became significantly more consistent.

In addition, the continuity assumptions on the measure make it impossible to use the current framework to effectively learn a version of the copy task where the predicted output is required to be placed in order. However, on this task, we observe that the Mamba and transformer architectures perform exceptionally well.

