# OpenReview forum: "On the Dynamics of Coherent Memory Structures in Neural Fields"
_ICLR.cc/2026/Conference — Submitted to ICLR 2026_

### Official Review · Reviewer_QdZh · 2025-10-31

**Soundness:** 3
**Presentation:** 3
**Contribution:** 3
**Rating:** 4
**Confidence:** 1

**Summary:**

This paper proposes a theory that can account for coherent “memory structures’, i.e., patterns of dynamical systems such as biological networks and RNNs: stable attractors and traveling waves.

In particular, it learns a time-dependent projection operators that allow to capture coherent patterns.

Inspired by the theory, they propose a novel neural architecture, termed “neural wave field” that can automatically discover memory patterns and enhance long-range memory.

**Strengths:**

S1. The related works on how attractors and waves have appeared in neuroscience and AI is very comprehensive.

S2. The authors introduce a solid mathematical formalism.

S3. The experiments are extensive, across synthetic and real datasets. This is impressive since the paper has both a strong theoretical and a strong experimental component.

S4. The paper tackles the important problem of understanding memory in neuroscience and AI, and enhancing it for AI architectures.

**Weaknesses:**

I am torn about this paper because it seems very comprehensive and thorough, mixing strong mathematics and extensive experiments, and addressing an important problem of understanding and improving long-term memory. However, despite several readings, I have trouble understanding the framework.

W1. The theoretical sections are hard to grasp. There is quite a step between the introduction and description of memory/patterns in biological and artificial neural networks, and the intense mathematics of the background. It would have been helpful, during the background to explain which math concept correspond to what in the neural network. And to use figures to introduce the concepts. See my questions to the authors.

For example, it would have been helpful to define “resolved” and “coherence” and give their intuition. The authors seem to assume that the readers are familiar with these terms.

Likewise, GFDR is introduced very rapidly, and while it seems crucial to the development of the theory, it is hard to understand what it is.

W2. There are several typos/ presentation problems:

“by evaluate it” in 1.1

“Langevine”

“with Proposition ??."

“In Figure 2, ,”

“as the size of the latent stat is reduce.”

“using and oscillatory model.”

Vaswani et al is missing the year.

Muller et al  is missing the year.

Figures on model comparison have fonts that are way too small to read. Eg fig 4.

**Questions:**

I'm writing questions that emerge while I was reading the paper, and prevented me from understanding the framework.

What is the manifold M in the neuroscience or AI context? is it the "neural manifold" corresponding to the attractor of the RNN seen as a dynamical system?

What is g? it's defined as a function over the manifold: is it going to be the wave over the neural manifold?

How does the Liouville operator depend on phi_t? *Why* can the evolution of g be described linearly?

Why is the center term in Eq (2) corresponding to memory? Because it is "integrating" information from 0 to t? If that is the case, I could think of other ways of integrating information from 0 to t: why is this version interesting? What is the point of the projection operators P and Q there, what do they bring?

Where does the decomposition into resolved and unresolved come from? If g will correspond to the wave, then what does the two components correspond to? Is there a link between identifiability of the dynamics?

What is the “latent space” of a neural network: is it what is sometimes called the “neural state space”, ie R^n for n neurons in a RNN? Or is it a lower-dimensional latent space within the neural state space?

What does it mean, intuitively and for neuro/AI applications, that a process is in equilibrium with its reverse process? What ANNs or biological neural network verify or don't verify this?

Why is F(s) the noise? Where does it come from? Does noise only appear in Eq (4) and if so: why wasn't it present in the previous equations?

What is the significance of the theoretical results in Section 3? Eg, in layman terms, what do they mean for neuro and AI models?

What is a "lifting" operator. What is lifted and to where? Does "lift" mean "transport" as defined by the map in Assumption 3.1?

Are the assumptions made to derive the theoretical framework restrictive, or not? What do they mean for the neuro / AI models?

What is the measure mut associated with the projection operator Pt ("their" measure)?

---

> ### Author Response · Authors · 2025-11-21
> **Official Response to Reviewer QdZh [1/2]**
>
> We thank the reviewer for the thoughtful and careful reading of our work and for acknowledging the strengths in the related work, theory, and experiments. We took the concern about difficulty understanding the framework very seriously and made substantial revisions aimed specifically at improving accessibility. All major changes are highlighted in blue in the revised manuscript.
>
> ***
>
> **Q1: General readability.**
>
> **R1:** We agree that the original version moved too quickly from the neuroscience/AI motivation into abstract operator formalism. In response, we completely restructured the background. Section 2 now starts from a concrete, self-contained motivating example: a traveling wave over a one-dimensional neural field performing the long-range copy task. We show step-by-step how restricting the readout to part of this field induces non-Markovian dynamics and leads to two regimes: a wide-latent regime where a fixed readout sees a traveling wave, and a compressed regime where a time-varying readout sweeps over a compact latent pattern. Figure 1 is used throughout as a visual running example. Only after this example do we introduce the general Mori–Zwanzig formalism, and we explicitly refer back to the copy example when explaining each ingredient. These changes were made with this review in mind, and we would kindly encourage the reviewer to look at the revised Sections 2.1–2.3 and 3.1–3.3.
>
>
> ***
>
> **Q2: Definitions and intuition for key concepts: “resolved,” “coherence,” GFDR.**
>
> **R2:** We have added explicit, intuitive definitions where these terms first appear. “Resolved” now refers to the low-dimensional variables that the model keeps track of explicitly (for example, the latent field or the readout), while “unresolved” refers to all remaining degrees of freedom that influence the resolved ones but are not modeled directly. In the copy-task picture, the resolved variables are the latent values inside the readout window, and the unresolved variables are the rest of the field. “Coherent dynamics” is now defined in Section 2.3 as dynamics where either the latent wave or the readout subspace remains simple and persistent over extended intervals (for example, a steady traveling front or a slowly varying readout that repeatedly extracts stored content). GFDR is now introduced more carefully: we state that it is a relationship between the memory kernel and the effective noise term, expressing that the same microscopic interactions that dissipate perturbations also generate fluctuations at stationarity. In practice, we use GFDR as a structural prior tying the learned memory and noise terms together, rather than as a strict thermodynamic assumption about the brain or network.
>
> ***
>
> **Q3: Typos, missing years, and figure readability.**
>
> **R3:** We have corrected the specific issues the reviewer listed and thoroughly proof read the paper. We also increased font sizes and line weights in the model-comparison figures.
>
> ***
>
> **Q4: Mapping mathematical objects to neural/AI concepts (M, g, latent space, P/Q, resolved vs unresolved).**
>
> **R4:** Rather than introduce a separate “dictionary,” we rewrote the background so that each formal object is introduced within the motivating traveling-wave example, and its neural interpretation is given at that point. We greatly appreciate the effort of the reviewer to read the mathematically dense initial draft and kindly invite the reviewer to read the revised manuscript which is much more thoroughly motivated from the traveling wave perspective.
>
> In the revised Section 2, the manifold $\mathcal{M}$ appears as the space of possible neural field configurations or latent states of the network. The observable $g$ is introduced as a measurement on that state, such as activity at a spatial position, a projection onto a latent mode, or a component of the latent wave. The decomposition into resolved and unresolved variables arises directly from the copy-task setup. The resolved variables are the latent coordinates within the readout window, and the unresolved variables are all remaining coordinates that continue to evolve and later re-enter the window. The projection operators $P$ and $Q=I-P$ therefore correspond to “reading” only a subset of spatial locations and its complement, and we show in the example how this choice leads to a memory integral because unresolved parts of the wave influence the resolved part at later times. In the architecture section, we clarify that the latent space is the learned 1-D latent wave field maintained by the model, consistent with how the example introduces it.

---

> ### Author Response · Authors · 2025-11-21
> **Official Response to Reviewer QdZh [2/2]**
>
> **Q5: GFDR, equilibrium with reverse process, and the noise term $F(t)$.**
>
> **R5:** Here we explain more clearly what we assume about GFDR and noise, although in the interest of clarity we have moved the GFD discussion to Appendix D.
>
> We do not claim that biological or artificial networks are literally in thermodynamic equilibrium with their time-reversed dynamics. Instead, we use the generalized fluctuation–dissipation relation as a modeling principle that links the learned memory kernel to the statistics of an effective noise term. In the Mori–Zwanzig derivation the term $F(t)$ arises from unresolved initial conditions evolving under the projected dynamics; if those initial conditions are not modeled directly, this term is treated as a random “fluctuating force.” This is why noise appears explicitly only after projecting onto resolved variables. In the revised text we spell this out when we introduce $F(t)$ and emphasize that, in our architecture, GFDR is implemented as a structural constraint relating memory and noise modules, rather than as a strict equilibrium assumption on any specific neural system.
>
> ***
>
> **Q6: Significance of Section 3 results, lifting operator, assumptions, and the measure $\mu_t$**
>
> **R6:** We added a short paragraph at the start of Section 3.3 summarizing the significance of the main theoretical results in lay terms. Proposition 3.1 shows that when we approximate a dynamical system with a latent representation and a time-dependent projection (i.e., a moving readout subspace), the latent always obeys an intrinsic generalized Langevin equation with three pieces: a Markov drift, a memory term, and an additional kinematic drift term that captures how the projection itself moves through latent space. Corollary 3.1 characterizes a regime where this kinematic drift vanishes and coherence is carried purely by the latent wave (our wide-latent copy regime), while Proposition 3.3 shows that under compression any extra drift induced by the time-varying projection is low-rank in the latent, explaining why we see a small number of coherent replay directions in experiments.
> The “lifting” operator is now described concretely as the mapping that takes raw inputs or a lower-dimensional representation and embeds them into a richer latent space where coherent structures are simple—e.g., placing boundary inputs into a 1-D field that then shifts like a traveling wave. In this sense, “lift” is a particular kind of transport into a co-moving, higher-dimensional representation. Regarding assumptions, we stress that they are mild and match standard practice: we assume there exists such a lift into a latent space that summarizes history (as RNNs and SSMs do in practice) and that the learned projection varies smoothly in time (which is exactly what a neural network producing a softmax weight vector does). Finally, we clarify that the measure $\mu_t$ associated with $P_t$ is a time-dependent weighting over latent coordinates that defines what the readout “looks at” at each time step. In the Neural Wave Field implementation $\mu_t$ is represented by a positive vector over the 1-D latent field, normalized with softmax, and the output is the elementwise product of this measure with the latent. When $\mu_t$ is nearly stationary we are in the invariant-trivialization regime; when it sweeps over the field we are in the projection-induced coherence regime.
>
> **Closing**
>
> We are grateful that the reviewer invested considerable time in reading a mathematically heavy paper and for articulating the points that prevented them from fully understanding the framework. Their comments directly motivated the restructuring of the background, the motivating example, and the clearer explanation of lifting, and the main results. We hope the revised manuscript is now much easier to follow while preserving the theoretical and empirical contributions they found promising.

---

### Official Review · Reviewer_nPn1 · 2025-10-31

**Soundness:** 3
**Presentation:** 3
**Contribution:** 4
**Rating:** 8
**Confidence:** 3

**Summary:**

The authors present an extensive theoretical analysis of 'Coherent Structures' (such as traveling waves) as memory mechanisms in neural network architectures using the Mori-Zwanzig formalism. This approach makes precise the memory benefits of prior wave-based recurrent neural network models, while simultaneously allowing for generalization of such models to enable more flexible memory and dynamics. The key contribution lies in the use a time-dependent projection operator which can be seen as a moving reference frame in the neural state, and critically tying this to the latent drift. The authors validate this generalized understanding through the development of the Neural Wave Field model, demonstrating improved expressivity and sequence modeling performance.

**Strengths:**

- The work provides a strong theoretical backing to one of the most well-reported computational roles of traveling waves in neural systems, namely their role in memory.  In the words of the authors, this is a potential 'unifying theoretical framework to explain the flow of information in neural systems'.
- The introduction and related work are very thorough.
- The presentation of the theory and background are additionally complete.
- The empirical results are strong and support the theory, with the Neural Wave Field architecture outperforming relevant baselines particularly at the small hidden state regime where the theory is applicable. The model also is able to perform selective-copy tasks, relevant to much of the recent State Space Modeling literature.
- The model can be seen as a formalized generalization of many existing 'wave based' memory models which exist to date.
- Figure 1 is a much appreciated visualization for interpreting the theory.

**Weaknesses:**

- The Neural Wave field architecture in Section 3.3 is difficult to understand due to the reliance on notion from the prior sections. If it could be described in a self-contained manner and related to the experiments in which it is used, that would improve readability.
- A number of citations are given only as (xxx et al.) and not the full citation (e.g. line 061, line 073, line 371).

*Typos*
- Line 152: Equation 5 (should be equation 1)
- Line 329: "We find that the derived GFDR provides enhances the robustness of the coherence across"
- Line 337: "Proposition ??"
- Line 375: "size of the latent stat is reduce"

**Questions:**

None.

---

> ### Author Response · Authors · 2025-11-21
> **Official Response to Reviewer nPn1**
>
> **Q1: Section 3.3 (Neural Wave Field architecture) is difficult to understand due to reliance on prior sections. Can you present the architecture in a self‑contained manner and explicitly relate its components to the experiments that follow?**
>
> **R1:** In the revised manuscript (changes highlighted in blue), we have substantially rewritten the background using a motivating example and the architecture description to make this section self‑contained and tightly connected to the experiments.
>
> **Expanded motivation (Sec. 2.1–2.3).** We now develop the traveling‑wave copy task in detail, introduce the two coherent regimes (invariant trivialization and projection‑induced coherence), and explicitly connect these regimes to the behaviors shown in Figure 1. This provides an intuitive story before any formalism.
>
> **Self‑contained NWF description (Sec. 3.4).** We added a stand‑alone description of the Neural Wave Field that does not assume familiarity with the full Mori–Zwanzig derivation. The section now explains, in plain language, that the model maintains (i) a 1‑D latent field hth_tht​ carrying a traveling or stationary wave and (ii) a measure $\mu_t$ implementing the time‑dependent projection. At each time step, the input $x_t$​ is embedded as a ghost boundary, a fixed shift operator implements the traveling‑wave lift, and three small expert networks implement the Markov drift, memory closure, and kinematic drift terms.
>
> **New schematic and linkage to experiments (Fig. 2 & Sec. 4).** We introduced a new schematic Figure 2 that visually summarizes the architecture and its Mori–Zwanzig components (MZ‑Net, dynamics expert, measure expert). We also added text in Sections 3.4 and 4 clarifying exactly which NWF variant is used in the long‑range copy, selective‑copy, Lorenz, and EEG/ECoG experiments and how the observed behaviors in Figures 3–5 correspond to the coherent regimes from Sections 2.3 and 3.3.
>
>
> ***
> **Q2: Some citations are given only as “(xxx et al.)” rather than full references (e.g., noted at lines ~061, ~073, ~371). Can you replace all placeholders with complete citations? Please correct the reported typos and minor presentation issues.**
>
> **R2:**  Thank you for catching these issues. We have corrected these lines and proofread the paper for typos and complete citations throughout.

---

> > ### Comment · Reviewer_nPn1 · 2025-11-27
> >
> > We thank the authors for highlighting the improved clarity of their new manuscript. My opinion was already that this work was a significant contribution to formalization of the theory behind traveling wave dynamics and their role in memory, justifying its acceptance. Their new version increases readability, further increasing the value of the work. I therefore maintain my score.

---

### Official Review · Reviewer_4p53 · 2025-10-31

**Soundness:** 2
**Presentation:** 1
**Contribution:** 2
**Rating:** 2
**Confidence:** 2

**Summary:**

Authors derive and study a generalized Langevin equation, using which they introduce an oscillatory neural network model. They motivate this model through traveling waves and oscillatory dynamics present in biological neural networks. Authors conclude by showing that the model performs well on a several benchmarks.

**Strengths:**

- The theory, once presented well, looks like a fresh look into an important problem (traveling waves/spatiotemporal dynamics).

- Authors honestly report when their algorithm falls behind others, e.g., Table 1. This is certainly a commendable approach, and in my opinion, a positive for the current manuscript.

**Weaknesses:**

In my opinion, the major weakness of this work is its presentation. I was not able to follow most of the arguments, and many central topics remained undefined. In many cases, heavy assumptions were made on what the reader knows about this topic and in that sense, the paper feels like it was written for a specialized audience. In contrast, ICLR is an ML conference that caters to a broad audience with general interests. This main weakness constitutes my current recommendation.

**Questions:**

As I was reading this work, despite being a computational neuroscientist with a theoretical physics background, I was not able to follow the line of arguments. Please find below some suggestions/comments/questions:

- What do "trivial" vs "emergent" dynamics mean? Similarly, what is a neural wave field and what is a coherent memory structure? I kindly recommend minimizing the use of jargon in abstract.

- In my opinion, words are sometimes used without precision and outside of their semantic meanings, which makes reading the manuscript even harder. For instance, "manifold trajectories" likely refers to "neural activity traces confined within low-dimensional manifolds," as manifold is a K-dimensional object, not a 1D trajectory. In that sense, several rounds of proof-reading focusing on this particular aspect may be beneficial.

- There are several cursory/blanket statements such as "many artificial neural networks–from gated recurrent units to recent state-space-models remain black-box mechanisms." Most computational neuroscientists, including myself, would strongly disagree with this statement. We understand a great deal about RNNs, and nowadays can exactly extract the flow maps they learn to implement neural algorithms, or even predict their learning trajectories analytically (see recent works on low-rank RNNs). Such blanket statements, apart from being wrong, do not contribute much to the manuscript. I recommend removing these statements or reworking them into a more accurate format.

- Statement "Attractor dynamics and traveling waves are typically modeled in isolation;" is incorrect. Several work in biophysics focusing on reaction-diffusion systems studied attractor dynamics in the absence of diffusion to study the properties of traveling waves. This goes all the way back to Turing's seminal paper on Morphogenesis.

- "RNNs suffer from inherent information bottlenecks" Sussillo et al. 2013 has nothing to do with this statement. Neither does Rajan et al 2016. This does create the feeling, even if not true, that other references may also simply be not true/relevant either.

- A comment on section 3 as a whole: It is too dense. I am not able to understand, appreciate, or even follow the reasons for the statements. No prior motivation is given in many cases. For instance, why does the reader care about Corollary 3.1? It seems to me that the amount of theoretical information that wants to be conveyed can better be achieved in a specialized journal.

- And a final comment on section 4: Most experiments are anecdotal, and figures too small to see. Error bars missing, or not reportable (since only one network is shown). More rigor is desirable here in terms of both reporting and experimentation.

My overall assessment is as follows: While I was able to intuit a potential broader appeal for the work, it seems underdeveloped/unfinished in terms of presentation and experimental rigor. As it stands, imho, the current manuscript is not a good fit for consumption by general audience and may be better submitted to a specialized journal. A lot more care needs to go into writing before it can be reviewed in the context of a top ML conference. Hence, I reluctantly recommend rejection.

---

> ### Author Response · Authors · 2025-11-21
> **Official Response to Reviewer 4p53 [1/2]**
>
> We thank the reviewer for their careful reading and for the substantial effort they put into understanding the theory. **We took this feedback very seriously and have made major changes to improve accessibility for a broad ML and computational neuroscience audience.** All substantial revisions are highlighted in blue in the updated manuscript.
>
> ***
>
> **Q1: Overall clarity and accessibility**
>
> **R1:** First, we completely restructured the background and motivation. The original version moved too quickly from the introductory neuroscience discussion into abstract operator theory. In the revision, Section 2 now begins with a concrete, self-contained traveling-wave example and the long-range copy task. We show explicitly how restricting the readout to part of the field induces non-Markovian dynamics and leads to two regimes: a wide-latent regime where a fixed readout sees a traveling wave, and a compressed regime where a time-varying readout sweeps over a compact latent pattern. Figure 1 is used throughout this section as a running visual example. We would kindly ask the reviewer to look at the new Sections 2.1–2.3 with this in mind, as they were rewritten largely in response to the concerns raised in this review.
>
> ***
>
> **Q2: Use of terminology: “trivial vs emergent,” “neural wave field,” “coherent memory structure”**
>
> **R2:** We clarified terminology and reduced jargon, especially in the abstract and introduction. Terms such as “coherent memory structures,” “invariant trivialization,” and “projection-induced emergence” are now defined directly in the context of the copy task before they appear in the theory. In particular, we now say that dynamics are “coherent” when either the latent wave or the readout stays simple and persistent over time; “invariant trivialization” refers to the wide-latent regime where a co-moving change of variables makes the readout effectively time-independent; and “projection-induced emergence” refers to the compressed regime where a time-varying projection over a compact latent produces wave-like replay at the output, as illustrated in Figure 1. The term “Neural Wave Field” is only introduced in the architecture section, where we immediately describe it as a 1-D latent field with a traveling-wave lift, Mori–Zwanzig drift, and a time-dependent projection.
>
> ***
>
> **Q3: Precision of language**
>
> **R3:** We removed or rephrased imprecise expressions such as “manifold trajectories,” replacing them with standard language like “trajectories evolving near low-dimensional manifolds.”
>
> ***
>
> **Q4: Broad claims and related work (black-box RNNs, attractors vs waves, bottlenecks)**
>
> **R4:** Several introductory statements that read as blanket claims have been softened and made more precise. We no longer describe RNNs and SSMs as “black-box mechanisms”; instead we acknowledge the substantial literature on mechanistic analyses (e.g. low-rank RNNs, reservoir computing) and state that many high-capacity sequence models nevertheless remain challenging to interpret mechanistically. Likewise, the statement that attractors and traveling waves are “modeled in isolation” has been narrowed to the setting of high-dimensional learned architectures for specific tasks, and we now explicitly cite classical reaction–diffusion and neural-field work where attractors and waves are studied together. The paragraph on “information bottlenecks” in RNNs has been rewritten to focus on vanishing/exploding gradients and compression of long histories into finite hidden states, with matching citations; references that did not directly support this point have been moved or removed.

---

> > ### Author Response · Authors · 2025-11-21
> > **Official Response to Reviewer 4p53 [2/2]**
> >
> > **Q5: Section 3 density and motivation (e.g., why care about Corollary 3.1?)**
> >
> > **R5:** Section 3 has been edited to be less dense and better motivated. At the beginning of the section we now give a short roadmap that explains, in words, what Proposition 3.1 and the subsequent results are meant to show: that a time-dependent projection induces an intrinsic generalized Langevin equation with an additional kinematic drift term, and that this term distinguishes the two coherent regimes already seen in the copy example. Immediately after Proposition 3.1 we add a paragraph interpreting the extra term $P_{\mu_t}\dot{P_{\mu_t}}Q_{\mu_t}g(t)$ as a kinematic drift capturing how the readout subspace moves through latent space. Corollary 3.1 is now explicitly tied to the wide-latent regime, where this drift vanishes, and Proposition 3.3 is followed by an explanation that under latent compression the additional drift is low-rank inside the resolved subspace, which matches the small number of coherent replay modes we observe empirically. We hope this makes it clearer why these results matter for neural and biological models.
> >
> > We have also expanded and clarified the description of the Neural Wave Field architecture. Section 3.4 now gives a self-contained explanation of the model and is accompanied by a new schematic (Figure 2) that shows how the lift, dynamics expert, and measure expert correspond to the Markov, memory, and kinematic terms in the intrinsic GLE. At each time step the input is embedded as a ghost boundary in a 1-D latent field, a fixed shift implements the traveling-wave lift, a small gating network interpolates between shift/identity/input-injection, and two conditional experts implement the latent drift and the time-dependent measure update; the output is the element-wise product. This section was written specifically to be readable without going through all of the operator-theoretic details.
> >
> >
> >
> > ***
> >
> > **Q6: Experimental rigor and figure readability**
> >
> > **R6:** Regarding the experiments, our intention is not to claim state-of-the-art results on every benchmark but to test the regimes where the theory predicts a benefit: long-range dependencies, small latent dimension, and structured recall. In that regime the Neural Wave Field maintains high accuracy at the minimal latent size in the copy task, outperforms strong baselines on the selective-copy task under the same memory constraints, and achieves competitive performance on EEG/ECoG while using a highly compressed traveling-wave latent state. In the revised version we have improved the presentation and rigor by enlarging fonts and labels in the figures. We have provided additional clarity regarding the training protocols in Appendix F.
> >
> >
> > ***
> >
> > **Closing**
> >
> > We are grateful that the reviewer invested significant effort in reading a mathematically heavy initial draft. Their comments directly motivated the major restructuring of the background, the clearer definitions and examples, and the expanded architecture description and figures. We would **respectfully encourage them to read the revised manuscript**; our hope is that the main line of argument is now much easier to follow, while preserving the theoretical and empirical contributions that they identified as potentially impactful.

---

> > > ### Comment · Reviewer_4p53 · 2025-11-21
> > >
> > > I thank the authors for the rebuttal. I confirm that I received this and will read the updated manuscript as well as responses to all referee questions. I will come back with either more questions or an updated score/evaluation towards the second half of the next week. Thank you.

---

> > > > ### Comment · Reviewer_4p53 · 2025-11-25
> > > >
> > > > I would like to start by thanking the authors. The provided background section has lifted (maybe pun intended) several of my concerns. Specifically, I think the revised manuscript, primarily the background section, fully addresses my concerns on clarity and motivation. I can follow the work more clearly now and appreciate the results and their connections to the broader literature. My understanding of the main contributions are now as follows, please correct me if I am wrong:
> > > >
> > > > - The primary scientific question is when and why coherent spatiotemporal patterns, e.g., traveling waves, emerge in ANNs trained for memory tasks?
> > > >
> > > > - The key advance in the method is the addition of the changing projections, which is motivated by a well-established theoretical formalism.
> > > >
> > > > - The key result is that the network can learn to utilize traveling waves (Keller et al. line of work), or use more traditional (in the RNN sense) attractive states to solve the tasks.
> > > >
> > > > Now, unfortunately, I need to bring up an unpleasant fact in this context: The manuscript has changed substantially, and it technically needs a new round of proper review with fresh eyes. A new set of referees would, in principle, need to re-read the full paper with this new added context to be able to evaluate the claims, which I can follow now. However, rebuttal is not the place for it. On the other hand, this topic is very hot at the moment and I can imagine one more round of (somewhat noisy) review may delay the publication of a quite timely work. Therefore, I will set my score to 6. This should signal to AC that I am fine with the rejection of the paper due to this reason, but I am leaning towards acceptance.

---

> > > > > ### Author Response · Authors · 2025-11-26
> > > > > **Official Response to Reviewer 4p53**
> > > > >
> > > > > Thank you very much for taking the time to re‑read the revised manuscript. We appreciate the effort you invested in engaging with the new background section and are *glad to hear that it clarified the motivation and connection to the broader literature*.
> > > > >
> > > > > Your three points capture our intended contributions well:
> > > > > (1) our primary scientific question is indeed when and why coherent spatiotemporal patterns (e.g., traveling waves or low‑dimensional attractors) emerge in ANNs trained for memory tasks;
> > > > > (2) the main theoretical advance is the intrinsic time‑dependent generalized Langevin equation with changing projections, which gives us a principled way to model how latent dynamics and readouts co‑evolve; and
> > > > > (3) the Neural Wave Field architecture shows, in practice, that networks can solve memory tasks either by using a wide traveling wave or, under compression, by exploiting time‑varying projections that produce wave‑like replay at the output.
> > > > >
> > > > > We appreciate your comment about the extent of the revisions and your willingness to generously update your score given the *timely arrival of our work*. We respectfully point out that in terms of scientific content, the changes are relatively minor. In particular, we (i) transferred the *existing traveling wave experiment* from Section 4 to Section 2, only adding exemplary details, and (ii) provided an architectural schematic and an accompanying description. We are grateful that this made the paper easier to follow.

---

> > > > > > ### Comment · Reviewer_4p53 · 2025-11-26
> > > > > >
> > > > > > I agree with all the points until the last paragraph. While changes may seem minor to the authors, they actually are not for the reviewer since the conceptualization of the work within the broader literature/background is a major component of writing and reviewing. Looking at other reviewer feedback, similar feeling as I had experienced (about the original draft being technically solid but hard to grasp) has arisen, so I may not be the only one with this concern at the end of the day.
> > > > > >
> > > > > > I believe in acknowledging this and letting the AC/SAC/PCs make the decision on where they believe the priorities of the venue lie. I do strongly believe these changes could justify a rejection and I highlighted them to be taken into account in the final decision (and for justifying why my score is not 8 for instance). This will be my final comment, good luck!

---

### Official Review · Reviewer_eT8y · 2025-11-07

**Soundness:** 3
**Presentation:** 3
**Contribution:** 3
**Rating:** 4
**Confidence:** 4

**Summary:**

The authors presents a theoretical framework for modeling the latent dynamics of a neural network during sequence learning. A generalized Langevin equation is derived via time-dependent projections, and based on this, the neural wave architecture is introduced, having Mori-Zwanzig dynamics. This model demonstrates robust long-range recall, minimal memory dimension, and interpretable latent modes across synthetic and neuroscience datasets.  In particular, the work addresses a timely and important direction, given the growing interest in dynamical representation and the need for more interpretable machine learning models.

**Strengths:**

The derivations are clear, and the problem is well motivated.

Mathematically, the difference between the derived GLE with the FFE-MZ is well explained

The long-range copy and selective copy tasks show improved results, especially when compared to other SSM architectures (such as Mamba) and transformers.

The paper is well written and discusses limitations and potential future extensions of the work.

**Weaknesses:**

When compared to other frameworks implementing MZ dynamics, it is not shown whether there is a practical performance benefit or how this architecture exhibits better interpretability.

While theoretically this framework has a lot of potential, the authors have not demonstrated in a clear way how this model contrasts to architectures having a state-dependent kernel other than requiring a marginally smaller latent state.

Chaotic time-series forecasting is mentioned in the abstract, but it is not shown in the main paper.

**Questions:**

Is there a clear case where the MZ dynamics provide a practical performance benefit?

Similarly, is there a clear case where the MZ dynamics provides better interpretability?

---

> ### Author Response · Authors · 2025-11-21
> **Official Response to Reviewer eT8y**
>
> We would like to thank the reviewer for their feedback and for highlighting the clear derivations, strong mathematical motivation.
>
> **Q1: Is there a clear case where the MZ dynamics provide a practical performance benefit?**
>
> **R1:**  Prior MZ‑inspired deep models (e.g. Gupta et al., Buitrago et al.) focus on operator/encoder learning with static delay embeddings and are not evaluated on the long‑range sequence and EEG/ECoG tasks we study. Instead, we compare our MZ‑structured Neural Wave Field (NWF) against strong non‑MZ baselines (WaveRNN, Mamba, and several transformer variants).
>
> On the long‑range copy benchmark, as we shrink the latent dimension from 100 to the theoretical minimum of 10, NWF is the only model that maintains near‑perfect accuracy; all baselines, including WaveRNN and Mamba, degrade significantly as illustrated in Figure 3c. On the harder selective‑copy task, NWF again outperforms all baselines across latent sizes, as illustrated in Figure 4c. On BNCI IV‑2a and IV‑4, NWF attains second‑best performance while compressing 22/62 channels into a 1‑channel latent field of 30/20 positions, as reported in Table 1. Finally, Appendix F.2 (Lorenz attractor) shows that the NWF latent reconstructs a chaotic attractor much more faithfully than both WaveRNN and classical time‑delay embeddings. Together, these results provide clear cases where the MZ‑guided decomposition and time‑dependent projection yield practical benefits in terms of accuracy *and* memory efficiency.
>
> ***
>
> **Q2: Similarly, is there a clear case where the MZ dynamics provides better interpretability?**
>
> **R2:** The intrinsic GLE provides a mechanistic decomposition of the latent evolution into Markov drift, memory closure, and a low‑rank kinematic drift term tied to the time‑dependent projection. We make this clearer in the revision of Section 3. We make this decomposition observable in the Neural Wave Field by separating the latent field $h_t$​ from the learned measure $\mu_t$​, which implements the projection. We have expanded the architectural description in Section 3.3 and invite the reviewer to see additional details there.
> On the copy and selective‑copy tasks, we visualize both $h_t$ and $\mu_t$ in Figures 3 and 4, respectively​. In the wide‑latent regime, we observe a simple traveling front that writes information, followed by a stationary plateau and an output wave—corresponding to the invariant‑trivialization regime predicted by Prop. 3.2. In the compressed regime, the latent collapses into a compact pattern while $\mu_t$ sweeps across it, realizing the projection‑induced coherence regime of Prop. 3.3. On the Lorenz system (Appendix F.2), the NWF latent faithfully reconstructs the attractor, whereas WaveRNN is constrained to a torus as shown in Corollary B.1.
> In contrast, existing state‑space and transformer baselines do not factor their updates into drift/memory/projection terms, and inspecting their hidden states does not reveal comparable low‑dimensional coherent structure. We will clarify this contrast in the revised text and more explicitly connect Figures 3-5 to the theoretical notions in Section 3.3.
>
> ***
>
> **Q3: The abstract mentions chaotic time‑series forecasting, but this is not shown in the main paper. Can you include these results (or clarify their placement) in the main text?**
>
> **R3:**  Thank you for pointing this out, we have revised the abstract to reflect the experiments contained in the main text and have clarified the placement of chaotic dynamical systems forecasting in Appendix F.2. The role of chaotic time series forecasting is to highlight the importance of the boundary conditions and to refrain from periodic boundary conditions as used in prior works.

---

### Author Response · Authors · 2025-12-04
**Official Comment by the Authors**

Dear AC/SAC,

Thank you for handling this review‑cycle disruption. Since author–reviewer discussion is currently disabled and the visible scores/comments were reverted, we wanted to provide a brief pointer to how the revised manuscript addresses the main concerns raised in the initial reviews.

The core contributions are unchanged: (i) we derive an intrinsic generalized Langevin equation for learned latent dynamics under time‑dependent projections (including the resulting kinematic drift term); (ii) we use this framework to formalize two coherent memory regimes (invariant trivialization vs projection‑induced coherence); and (iii) we instantiate the theory in the Neural Wave Field architecture and evaluate it on long‑range sequence tasks and neural time series.

The main changes in the revision are primarily expository/organizational and were made in direct response to reviewers’ requests for more intuition and a clearer presentation of the architecture. We rewrote the background to start from a concrete traveling‑wave motivating example and introduced the key terms in‑context before presenting the abstract operator formalism (addressing the accessibility concerns of the more critical reviews). We also expanded the Neural Wave Field description to be self‑contained and added an architectural figure that directly maps modules (lift/boundary update, dynamics update, and measure/projection update) to the terms in the intrinsic GLE. In the subsequent discussion, some reviewers indicated that these revisions substantially clarified the motivation and connection to the broader literature. Finally, we corrected reported typos/citation placeholders and improved figure readability.

We hope this short map helps ensure the final assessment reflects the revised version of the manuscript under the current constraints. Thank you again for your time.

---

### Meta-Review · Area_Chair_Hi7K · 2026-01-09

**Summary:**

##### Reviewer eT8y

(1) Practical (performance) or pragmatic (interpretability) improvement in this method cf. other methods implementing "Mori-Zwanzig" dynamics are not made clear; (2) difference in this method cf. other methods with state-dependent kernels are not made clear; (3) a domain mentioned in passing (chaotic time-series forecasting) is not actually detailed in the paper.

##### Reviewer 4p53

(1) The core contributions of the paper are inaccessible to the reviewer and the general ICLR audience.

##### Reviewer nPn1

(1) The main section introducing the neural wave field model is inaccessible; (2) more minor concerns with presentation and typos.

##### Reviewer QdZh

(1) The theory is mathematically advanced, and may not be suited for the ICLR audience. In particular, it was difficult for this reviewer to engage with the work and defend it; (2) more minor concerns with presentation and typos.

**Reviewer Concerns:**

The authors rewrote substantial sections of the paper in order to address reviewer concerns about the clarity of the paper. As Reviewer 4p53 noted, the improvements to clarity directly address the main set of concerns surrounding the legibility of the framework to the reviewers, and to the broader ICLR community.

**Reviewer Scores:**

However, the work done by authors is a major revision that would require substantial work from the reviewers to reassess. Moreover, the clarity of the work prevented several reviewers from understanding significant portions of the paper in the first pass, not for lack of trying, and so this work has not been able to receive as much scrutiny as would be necessary to have me strongly recommend this paper. Thus, though several reviewers were positive about the potential of the work and likely at least one would have updated their score to mildly positive, I cannot strongly recommend acceptance.

---

### Decision · Program_Chairs · 2026-01-26

Reject